# Contrast-Aware Calibration for Fine-Tuned CLIP: Leveraging Image-Text Alignment

## Abstract

Vision-language models (VLMs), such as CLIP, have demonstrated exceptional generalization capabilities and can quickly adapt to downstream tasks through prompt tuning. Unfortunately, in classification tasks involving non-training classes, fine-tuned VLMs often overfit to train classes, resulting in a misalignment between confidence scores and actual accuracy on unseen classes, which significantly undermines their reliability in real-world deployments. Existing confidence calibration methods typically require training parameters or analyzing features from the training dataset, restricting their ability to generalize unseen classes without corresponding train data. Moreover, VLM-specific calibration methods rely solely on text features from train classes as calibration indicators, which inherently limits their ability to calibrate train classes and other evaluation settings, like cross-dataset and domain-generalization settings. To address these challenges, we propose a multimodal calibration method **Contrast-Aware Calibration (CAC)**. Building on the original CLIP's zero-shot adaptability and the conclusion from empirical analysis that poor intra-class and inter-class discriminative ability on unseen classes is the root cause, we calculate calibration weights based on the contrastive difference between the original and fine-tuned CLIP. This method is not only effective for calibrating unseen classes but also overcomes the limitations of previous VLM calibration methods that struggle to calibrate train classes and other settings. In multiple setting experiments with 5 fine-tuning methods, CAC achieves strong calibration in all settings without sacrificing accuracy.

## 1 Introduction

Vision-language models, such as CLIP (Radford et al., 2021), pre-trained on vast web-scale text-image datasets, have demonstrated impressive zero-shot capabilities and image-text alignment in downstream image classification tasks (Deng et al., 2009; Helber et al., 2019). Concurrently, various prompt learning methods for VLMs have been proposed to further enhance the performance of VLMs on specific tasks by leveraging a small amount of labeled data (Khattak et al., 2023a;b; Zhou et al., 2022b). Given CLIPs strong zero-shot adaptability, the open-vocabulary setting has become a standard for evaluating the performance of fine-tuned VLMs, where prompts are trained on a subset of classes and evaluated on both train and unseen classes (Lee et al., 2023; Tan et al., 2024).

Unfortunately, fine-tuned VLMs regularly overfit to train classes, forgetting the well-calibrated predictions and image-text alignment achieved during pre-training (Zhou et al., 2022a;b). For unseen classes, they often produce semantically unbalanced representations, leading to image-text misalignment and a significant discrepancy between confidence scores and actual accuracy (Guo et al., 2017a; Minderer et al., 2021). Existing calibration methods (Joy et al., 2023; Oh et al., 2023; Zadrozny & Elkan, 2001) typically rely on training or analyzing features from the training dataset, limiting their ability to calibrate classes outside the training dataset. Moreover, due to relying on text features from train classes and neglecting the critical characteristic of image-text alignment, the state-of-the-art (SOTA) VLMs' calibration method, Distance-Aware Calibration (Wang et al., 2024), struggles to calibrate train classes and other evaluation settings of fine-tuned VLMs, such as cross-dataset and domain-generalization settings.

To fundamentally address the miscalibration in fine-tuned CLIP, we conduct extensive empirical analysis, identifying poor intra-class and inter-class discriminative ability on unseen classes caused

by downstream task adaptation as the primary cause. Moreover, our experiments reveal that original CLIP trained on large-scale datasets tends to exhibit superior confidence calibration performance, consistent with the findings in Minderer et al. (2021); Tu et al. (2023). The above findings and the connection between the contrast metric and confidence calibration inspire us to develop **Contrast-Aware Calibration (CAC)** to achieve effective confidence calibration. CAC improves the intra-class and inter-class discriminative ability of fine-tuned CLIP by reweighting the logits using the well-aligned information from the original CLIP. Specifically, by leveraging the similarity between the logits generated by the original and fine-tuned CLIP, CAC realigns the image-text feature relationships of the fine-tuned CLIP across both seen and unseen classes, improving confidence calibration on all classes and enabling broader applicability across diverse task settings. As a post-hoc calibration technique, CAC does not affect the model's accuracy while delivering more reliable predictions.

Due to its design tailored for CLIP with strong image-text alignment, CAC can be applied to any set of classes and any evaluation setting of CLIP. To assess its confidence calibration effectiveness, we evaluated CAC on three settings (open-vocabulary, cross-dataset, and domain-generalization) across 11 datasets, applying it to 5 different prompt learning methods. Leveraging the image-text alignment of the original CLIP, CAC consistently achieves the best performance in confidence calibration, outperforming the current best calibration method MIR (Roelofs et al., 2022) and SOTA VLM-specific calibration method DAC (Wang et al., 2024). Notably, CAC demonstrates robust performance across various potential variations, including training sample shots, model backbones, datasets, fine-tuning methods, and evluation settings, highlighting its insensitivity to parameters.

In summary, the main contributions of this paper include:

- To investigate the issue of miscalibration in fine-tuned CLIP on unseen classes, we identify poor intra-class and inter-class discriminative ability as the root cause through empirical analysis, providing a reliable reference for future research.

- Benefiting from the inference mechanism of contrastive learning models, we establish a connection between the contrast metric and confidence calibration, enabling us to utilize image-text alignment techniques to address the incomplete calibration of previous traditional and single-modal calibration methods.

- Based on the analysis, we propose a novel method, Contrast-Aware Calibration, which addresses the limitations of previous methods, including suboptimal calibration, inability to handle all classes, and poor adaptability to diverse task settings.

- The experimental results on 11 datasets, testing the calibration of 5 fine-tuning methods, show that CAC outperforms existing calibration methods. In particular, CAC demonstrates strong robustness and parameter insensitivity.

## 2 RELATED WORKS

**Prompt Learning in Vision Language Models.** Due to the large parameter size of VLMs and the limited availability of training data for downstream tasks, it is impractical to fine-tune all parameters of the VLMs to adapt them to these tasks. To improve the generalization of the learnable language prompt (Zhou et al., 2022b; Zang et al., 2024; Mistretta et al., 2024), CoCoOp (Zhou et al., 2022a) generates a vision-conditional prompt by fusing the image feature and the learnable language prompts. KgCoOp (Yao et al., 2023) and ProGrad (Zhu et al., 2023) are other prompt-based methods for VLMs. MaPLe (Khattak et al., 2023a) and PromptSRC (Khattak et al., 2023b) conduct the visual-textual prompt learning by jointly conducting the prompt learning on the vision and text encoders. To address higher confidence bias for unseen classes caused by fine-tuning, the proposed method leverages the original CLIP to correct the confidence of fine-tuned CLIP.

**Confidence Calibration.** Confidence calibration aims to align the confidence scores predicted by models with their actual performance. A common strategy for achieving this is to apply calibration techniques after model training. These techniques can be broadly divided into two categories: scaling-based methods (Guo et al., 2017a; Xiong et al., 2023; Guo et al., 2017b) and bin-based methods (Roelofs et al., 2022; Zadrozny & Elkan, 2001; 2002). Distance-Aware Calibration (Wang et al., 2024) is the current SOTA calibration method specifically for VLMs; it estimates scaling weights for unseen class logits based on textual representations, but it performs poorly when class representations undergo substantial changes. As a newly proposed method, CAC enables effective

Table 1: Comparison of existing confidence calibration methods in terms of required information and applicable scenarios. "CG" represents the Calibration Granularity.

| Method | Train | Required Information | CG | Applicable Scenarios |
|--------|-------|---------------------|-----|---------------------|
| Bin Methods | No | Logits and ground-truth labels on training set | Dataset | train classes only |
| Scaling Methods | Yes | Logits and ground-truth labels on training set | Dataset | train classes only |
| DAC | No | Textual representations of train and unseen classes | Class | unseen classes within the same dataset |
| CAC (Ours) | No | Original and fine-tuned logits on testing data | Sample | train & unseen classes; cross-dataset; domain-generalization |

sample-level calibration across common experimental scenarios in VLM research, with a detailed comparison to existing methods provided in Table 1. In addition, some test-time adaptation methods have also discussed confidence (Yoon et al., 2024; Sharifdeen et al., 2025); however, since these methods affect the accuracy of the original model and differ substantially from calibration approaches, we do not consider them here.

**Calibation Evaluation Metrics.** A model is perfectly calibrated if $\mathbb{P}(\hat{y} = y | \hat{p} = p) = p$ for all $p$ in $[0, 1]$, where $y$ is the actual label, $\hat{y}$ the prediction, and $\hat{p}$ the confidence score. To assess model calibration, we typically use the Expected Calibration Error (ECE) (Guo et al., 2017a) , lower values indicating better calibration. ECE groups predictions into $M$ interval bins (each of size $1/M$ ) and calculate the accuracy of each bin. The ECE is defined as the difference between the accuracy and confidence of all bins, which can be calculated as: $\text{ECE} = \sum_{m=1}^{M} \frac{|B_m|}{M} |acc(B_m) - conf(B_m)|$. We also use other three metrics for evaluation: Maximum Calibration Error (MCE) (Guo et al., 2017a), Adaptive Calibration Error (ACE) (Nixon et al., 2019) and Proximity Informed Expected Calibration Error (PIECE) (Xiong et al., 2023).

## 3 ANALYSIS

In this section, we introduce the contrast metric and its calculation method, exploring the causes of poor confidence calibration in fine-tuned VLMs by examining feature representations of unseen classes. We then analyze the relationship between output logits and the contrast metric in contrastive learning models, which serves as the underlying logic for our method.

### 3.1 CONTRAST METRIC

Contrast is an indicator used to measure a models ability to distinguish between positive and negative samples, which is widely used in contrastive learning (Ko et al., 2022; Le-Khac et al., 2020). Models with strong discriminative capabilities typically exhibit higher contrast scores and vice versa. Specifically, given a similarity matrix $S \in \mathbb{R}^{N \times C}$, where $N$ represents the number of samples, $C$ denotes the number of classes, and $S[i, j]$ indicates the similarity between sample $i$ and class $j$, the contrast metric is computed through the following three components:

- **Positive Similarity.** For each sample $i$, the similarity score with its ground-truth label $y_i$ is extracted as $s_i^+ = S[i, y_i]$.

- **Negative Similarity.** For each sample $i$, the maximum similarity score among incorrect labels is calculated as $s_i^- = max_{j \neq y_i} S[i, j]$.

- **Difference Calculation.** The contrast metric is calculated as the difference between the average of positive and negative similarities:

$$\text{Contrast} = \frac{1}{N} \sum_{i=1}^{N} s_i^+ - \frac{1}{N} \sum_{i=1}^{N} s_i^- \tag{1}$$

For VLMs, contrast represents the models ability to distinguish between ground-truth and other classes, serving as an indicator of the models image-text alignment performance on datasets.

### 3.2 EMPIRICAL STUDY

Although prompt learning methods (Zhou et al., 2022a; Khattak et al., 2023a) freeze the parameters of the original CLIP, their learnable prompts often cause overfitting on train classes. For instance,

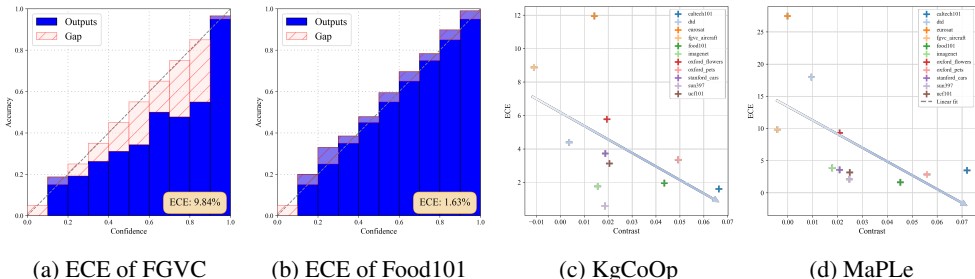

|            |            |            |            |
| :--------: | :--------: | :--------: | :--------: |
| (a) ECE of FGVC | (b) ECE of Food101 | (c) KgCoOp | (d) MaPLe |

Figure 1: (a) & (b) The reliability of KgCoOp evaluated on the FGVCAircraft and Food101 datasets. (c) & (d) The relationship between contrast and ECE for the logits output by KgCoOp and MaPLe across 11 datasets, with the lines representing the **negative correlation trend** among the variables.

KgCoOp (Yao et al., 2023) learns prompts tailored to train classes and ignores vision prompts, resulting in an imbalance between the text representations of unseen classes and the visual representations of the original CLIP, significantly diminishing its ability to accurately represent unseen classes and producing biased contrast scores (Khattak et al., 2023a;b; Wang et al., 2024). We observe that when the representations of fine-tuned VLMs deviate from the pre-trained imagetext alignment, *their class scores often become biased toward certain categories or converge to similar values across multiple classes*. This weakens the models discriminative ability between intra-class and inter-class samples established during pre-training, ultimately leading to miscalibration. Observations on FGVCAircraft (Maji et al., 2023) and Food101 (Bossard et al., 2014), which respectively represent high inter-class similarity and high intra-class variation, demonstrate that this phenomenon is the fundamental cause of fine-tuned VLMs miscalibration:

- **Observation 1: Overconfidence caused by inter-class similarity.** For FGVCAircraft, where inter-class similarity is high and class boundaries are hard to distinguish (Maji et al., 2023), fine-tuned VLMs tend to misclassify images into a few dominant classes, resulting in overconfident but incorrect predictions, as shown in Figure 1a. The contrast score results from untruth classes being more similar than ground-truth class, yielding a negative value. Such overconfidence is particularly evident in fine-grained classification tasks with overlapping inter-class features.

- **Observation 2: Underconfidence caused by intra-class variation.** For Food101, where inter-class similarity is low but significant intra-class variation exists (Bossard et al., 2014), fine-tuned VLMs often produce multiple high-confidence predictions across unseen classes, leading to lower overall confidence, as shown in Figure 1b. Despite this, the model can still identify the correct category, keeping the contrast score positive.

More detailed visualization and discussion about the above observations are shown in Appendix C. As presented in Figure 1c and Figure 1d, the results of KgCoOp and MaPLe show that higher contrast scores are generally associated with lower ECE values. Through extensive experimental analysis, we conclude that a models class discrimination ability directly impacts its calibration performance: **for unseen classes, contrast and ECE exhibit a negative correlation**, making contrast a reliable metric for adjusting ECE. Furthermore, the results indicate that **well-aligned vision-language models typically demonstrate better confidence calibration**, consistent with the observations reported in Minderer et al. (2021); Tu et al. (2023).

### 3.3 LOGITS AND CONTRAST

For VLMs like CLIP, which align visual and textual data in a shared embedding space using an image encoder $f$ and a text encoder $g$. Given a caption $T$, the text encoder produces features $g(E_w(T))$, where $E_w$ is the word embedding layer. Since these models are trained and make predictions based on contrastive learning, the similarity matrix $S$ used to calculate the contrast corresponds to the logits output by CLIP:

$$S = logits_{CLIP} = \frac{f(I) \cdot g(E_w(T))}{\|f(I)\|\|g(E_w(T))\|} \tag{2}$$

When the contrast is low, the logits from CLIP often indicate two or more classes with relatively high scores. Conversely, a higher contrast typically corresponds to a class with a significantly higher

score, while the scores for other classes remain much lower. This property is generally applicable to other CLIP-based models and helps establish the relationship between the logits output by VLMs and the contrast metric. By linking logits with the contrast metric, we turn our conclusions into a practical solution for contrastive learning models, providing the foundational logic for our method.

# 4 CONTRAST-AWARE CALIBRATION

In this section, we first introduce the fundamental **Contrast-Aware Calibration (CAC)** method, which is specifically designed for image-text well-aligned models based on the conclusions in Subsection 3.2. Next, we present an enhanced calibration method tailored for cases requiring stronger calibration, both of which are collectively referred to as CAC. Finally, we provide a brief overview of the CAC calibration process during inference and discuss its advantages.

## 4.1 CONTRAST-AWARE CALIBRATION

As demonstrated in Subsection 3.2 and existing studies (Minderer et al., 2021; Tu et al., 2023), the original CLIP, trained on large-scale image-text pairs, exhibits relatively high contrast across different datasets, leading to more conservative predictions compared to fine-tuned CLIP. This observation motivates us to leverage original CLIP as a reference to regulate the confident bias logits of fine-tuned CLIP. Specifically, we compute the $L_1$ distance between the logits of the original CLIP and fine-tuned CLIP, treating it as the confidence bias metric $z$. The formula is as follows:

$$z = \frac{1}{N} \sum_{i=1}^{N} |P_i - \hat{P}_i|, \tag{3}$$

where $N$ denotes the total number of classes, and $\hat{P} = \{\hat{p}_i\}_{i=1}^{N}$ and $P = \{p_i\}_{i=1}^{N}$ represent the logits output by the original and fine-tuned CLIP, respectively. In particular, according to our analysis, the logits of contrastive learning-based VLMs are equivalent to the contrast metric, so $z$ can serve as an indicator for measuring the confidence difference between the original and fine-tuned VLMs. To better leverage its negative correlation with ECE, we design the following function to transform $z$ into CAC weights:

$$\gamma = \alpha \cdot e^{-kz}. \tag{4}$$

This function addresses several issues caused by directly using $z$ as a confidence weight, such as reversed monotonic trends, minimal numerical differences, and an incomplete value range. The components of the function serve the following purposes:

- **Reason for choosing** $e^{-x}$: Due to the negative correlation between ECE and contrast metric, we choose a common decreasing function $e^{-x}$, which has a range of $[0, 1]$ and aligns with the required monotonicity.

- **Effect of** $k$: Since the text and image features in CLIP-based models undergo normalization before computing the logits, the $L_1$ distance may be small. Therefore, it is necessary to amplify the input to the function, with $k$ serving to scale $z$, and enable the function to capture input variations more effectively.

- **Effect of** $\alpha$: Since fine-tuned VLMs may be underconfidence and overconfident in various datasets, we modify the functions maximum value to $\alpha$ $(> 1)$, equipping CAC with the ability to deal with underconfidence.

In summary, the design of Equation 3 bridges the gap between contrast and the degree of miscalibration, providing a reliable metric to assess the CLIPs confidence calibration. By applying amplification and other operations, Equation 4 transforms $z$ into a suitable scaling factor, ultimately resulting in the CAC.

## 4.2 ADVANCED CONTRAST-AWARE CALIBRATION

The CAC proposed in the previous section can already be used as an effective calibration method for fine-tuned CLIP. However, different datasets and fine-tuning methods require varying levels of calibration. For example, KgCoOp requires stronger calibration, whereas PromptSRC typically

only needs minor adjustments. Therefore, we use an adaptive module to amplify the weights in underconfident cases and reduce them in overconfident cases, ultimately forming the CAC method:

$$\hat{\gamma} = \begin{cases} \gamma^2, & \text{if } \gamma < \lambda_1, \\ \gamma, & \text{if } \lambda_1 \leq \gamma \leq \lambda_2, \\ \gamma^2 & \text{if } \gamma > \lambda_2. \end{cases} \tag{5}$$

where $\hat{\gamma}$ represents the advanced CAC output, $\gamma$ represents the CAC weights, and $\lambda_1$ and $\lambda_2$ represent the boundary points of the interval for shrinking or amplifying $\gamma$. Compared to the original version, the Advanced CAC achieves more flexible confidence calibration through the integration of the following two parts:

- **Advantage of Piecewise Function**: When $\gamma$ is less than or greater than these thresholds, the model often exhibits significant underconfidence or overconfidence, respectively. Therefore, a piecewise function that adjusts the scaling weights for specific scenarios is necessary.

- **Advantage of Squaring**: For values within the range $[0, \alpha]$, squaring will offer the advantage of maintaining numerical stability without drastically affecting the values. Unlike scaling by a manually defined constant, this approach removes the need for hyperparameter fine-tuning, making it more efficient and reliable.

Ultimately, the proposed CAC is a robust confidence calibration method specifically tailored for CLIP with well-aligned image-text features. By scaling the logits of fine-tuned CLIP using those of the original CLIP, CAC achieves parameter efficiency, strong generalization capabilities, and a training-free implementation, addressing the limitations of previous methods that rely on training parameters or analyzing train data features.

## 4.3 CALIBRATED INFERENCE

Given an input image $i$, we first collect the CAC scores of this image, denoted as $\hat{\gamma}_i$, which is then used to calculate the rectified logits as follows:

$$L_i^{CAC} = \hat{\gamma}_i * \tau * logits_i \tag{6}$$

where $logits_i$ is the logit calculated by fine-tuned CLIP. As a post-hoc confidence calibration method, CAC exclusively scales the temperature coefficient $\tau$ of CLIP, improving the models reliability without compromising its original accuracy and inference speed. In summary, the advantages of our method are as follows:

- **Tailored for CLIP**: CAC is specifically designed for CLIP, leveraging the original CLIP's alignment between visual and textual modalities to achieve effective confidence calibration.

- **Simultaneous Scaling for Both Train and Unseen Classes**: As an improvement over previous methods, CAC calibrates the confidence of both train and unseen classes, addressing their limitation of only handling train or unseen classes.

- **Strong Empirical Foundation**: The design of CAC is grounded in experimental insights, ensuring its strong interpretability and reliability.

- **Simple Plug-and-Play**: CAC eliminates the need for additional training or extensive analysis of train data, providing a simple and efficient solution for enhancing confidence calibration.

## 5 EXPERIMENTS

### 5.1 EXPERIMENTAL SETUP

**Evaluation Paradigm.** Following the open-vocabulary setting in the VLM field (Khattak et al., 2023a; Zhou et al., 2022a), the datasets are divided into train and unseen classes. The model is trained on the train classes in a few-shot setting and we generally report the calibration performance on both train and unseen classes, which is different from DAC (Wang et al., 2024). Additionally, we evaluated the calibration performance of CAC in other VLM calibration settings, including cross-dataset and domain-generalization settings, where DAC is completely ineffective.

Table 2: Average ECE results across 11 datasets on train classes. Smaller values are better. "Conf" represents the original performance on open-vocabulary classes with existing tuning methods.

| Method | Conf | Training-free | | | | Training-based |
|---|---|---|---|---|---|---|
| | | IR | HB | MIR | CAC | TS |
| CoCoOp | $3.60_{\pm0.16}$ | $7.80_{\pm5.64}$ | $7.50_{\pm1.95}$ | $3.87_{\pm2.25}$ | $\mathbf{3.05}_{\pm0.54}$ | $3.42_{\pm5.01}$ |
| KgCoOp | $5.87_{\pm0.29}$ | $7.23_{\pm1.35}$ | $7.41_{\pm0.78}$ | $7.38_{\pm6.77}$ | $4.10_{\pm0.23}$ | $\mathbf{3.01}_{\pm2.35}$ |
| MaPLe | $2.80_{\pm0.50}$ | $7.81_{\pm7.08}$ | $6.63_{\pm1.72}$ | $2.70_{\pm0.50}$ | $\mathbf{2.50}_{\pm0.63}$ | $2.64_{\pm4.74}$ |
| ProGrad | $5.93_{\pm0.69}$ | $5.69_{\pm2.47}$ | $5.63_{\pm0.51}$ | $4.42_{\pm1.01}$ | $4.17_{\pm0.45}$ | $\mathbf{3.06}_{\pm2.71}$ |
| PromptSRC | $3.74_{\pm0.16}$ | $6.39_{\pm6.26}$ | $6.38_{\pm1.64}$ | $3.55_{\pm1.92}$ | $\mathbf{2.75}_{\pm0.22}$ | $2.86_{\pm3.25}$ |

**Compared Methods.** We mainly focus on benchmarking against the other 5 current representative prompt learning methods: CoCoOp (Zhou et al., 2022a), KgCoOp (Yao et al., 2023), MaPLe (Khattak et al., 2023a), ProGrad (Zhu et al., 2023), and PromptSRC (Khattak et al., 2023b). Since models like CoOp only consider the text modality and exhibit low accuracy, making their calibration significance minimal, we focused on testing and calibrating its optimized version, CoCoOp and KgCoOp. For train classes calibration, we select three representative calibration methods: Histogram Binning (HB) (Zadrozny & Elkan, 2001), Isotonic Regression (IR) (Zadrozny & Elkan, 2002), Multi-Isotonic Regression (MIR) (Roelofs et al., 2022), and Temperature Scaling (TS) (Guo et al., 2017b). For unseen classes calibration, we compare the SOTA method designed for CLIP, DAC (Wang et al., 2024). **Datasets** and **Evaluation Metrics** used in the experiments are shown in Appendix B.

**Implementation details.** For three settings, we use CLIP (ViT-B/16) as the pre-trained VLM throughout our experiments and report results of **Advanced CAC** averaged over 3 runs. In all experiments of this paper, we set $k = 15$ and $\alpha = 1.10$ as the default parameters for CAC. We select $\lambda_1 = 0.9$ and $\lambda_2 = 1.0$ as the threshold values for the piecewise function and compare the effects of this function in the ablation studies. Notably, all experiments were conducted using a single set of default parameters without the need for extensive fine-tuning. **Additional details** about pre-trained models, hyperparameters, and implementation specifics are provided in Appendix B.

## 5.2 EMPIRICAL RESULTS

**RQ1**: Compared with traditional calibration methods, can CAC achieve performance improvements on the train classes?

We respectively compared CAC with two types of calibration methods in Table 2:

- Compared with traditional training-free calibration methods, CAC consistently outperforms all of them, demonstrating its strong ability to calibrate VLMs. Notably, these methods often exhibit significant performance fluctuations across different fine-tuning methods, sometimes even increasing ECE rather than reducing it as intended.

- Compared with the training-based method (temperature scaling), which learns calibration coefficients tailored to individual datasets it exhibits superior performance under certain prompt learning methods. Nevertheless, CAC delivers a comparable level of effectiveness while preserving its training-free advantage.

Overall, compared with traditional calibration techniques, CAC exhibits strong effectiveness, achieving performance on par with methods that require additional training.

**RQ2**: What advantages does CAC offer over the current SOTA method?

DAC calibrates based on the high-dimensional distance between the textual representations of train and unseen classes, which restricts its applicability to unseen classes within the same dataset. As shown in Table 3, CAC achieves the best calibration performance even in this setting. Specifically, CAC demonstrates significant improvements on KgCoOp and ProGrad, while DAC tends to exhibit the opposite trend in performance optimization. Moreover, building on the well-calibrated foundation of the original CLIP design, CAC demonstrates stable results and robust performance across various settings when using default parameters. For **cross-dataset** (where class representations change significantly) and **domain-generalization** settings (where the classes remain unchanged) settings,

Table 3: Average calibration performance across 11 datasets on unseen classes.

| Method | ECE(↓) | | | ACE(↓) | | | MCE(↓) | | | PIECE(↓) | | |
|---|---|---|---|---|---|---|---|---|---|---|---|---|
| | Conf | DAC | CAC | Conf | DAC | CAC | Conf | DAC | CAC | Conf | DAC | CAC |
| CoCoOp | $5.44_{\pm1.65}$ | $5.70_{\pm2.14}$ | $\mathbf{4.24}_{\pm0.90}$ | $5.35_{\pm1.53}$ | $5.60_{\pm2.20}$ | $\mathbf{4.22}_{\pm0.95}$ | $1.38_{\pm0.12}$ | $1.40_{\pm0.46}$ | $\mathbf{1.20}_{\pm0.06}$ | $7.35_{\pm1.12}$ | $8.06_{\pm0.89}$ | $\mathbf{6.83}_{\pm0.49}$ |
| KgCoOp | $3.98_{\pm1.45}$ | $4.11_{\pm0.62}$ | $\mathbf{3.85}_{\pm1.55}$ | $3.93_{\pm1.59}$ | $4.09_{\pm0.58}$ | $\mathbf{3.78}_{\pm1.70}$ | $1.08_{\pm0.17}$ | $1.18_{\pm0.35}$ | $\mathbf{1.10}_{\pm0.28}$ | $6.45_{\pm0.84}$ | $6.62_{\pm0.48}$ | $\mathbf{6.39}_{\pm0.94}$ |
| MaPLe | $7.80_{\pm13.04}$ | $5.91_{\pm5.97}$ | $\mathbf{5.35}_{\pm8.26}$ | $7.77_{\pm12.80}$ | $5.93_{\pm5.95}$ | $\mathbf{5.30}_{\pm8.57}$ | $2.08_{\pm1.68}$ | $1.62_{\pm0.43}$ | $\mathbf{1.61}_{\pm1.60}$ | $9.53_{\pm8.06}$ | $8.19_{\pm5.04}$ | $\mathbf{7.69}_{\pm4.87}$ |
| ProGrad | $5.04_{\pm1.29}$ | $6.13_{\pm2.10}$ | $\mathbf{4.04}_{\pm1.43}$ | $4.95_{\pm1.37}$ | $6.18_{\pm2.44}$ | $\mathbf{4.05}_{\pm1.68}$ | $1.47_{\pm0.27}$ | $1.53_{\pm0.08}$ | $\mathbf{1.24}_{\pm0.10}$ | $7.41_{\pm1.20}$ | $8.20_{\pm0.98}$ | $\mathbf{6.75}_{\pm1.14}$ |
| PromptSRC | $4.29_{\pm4.14}$ | $4.55_{\pm4.68}$ | $\mathbf{3.47}_{\pm1.18}$ | $4.24_{\pm4.22}$ | $4.41_{\pm4.68}$ | $\mathbf{3.40}_{\pm1.18}$ | $1.16_{\pm0.28}$ | $1.17_{\pm0.38}$ | $\mathbf{1.03}_{\pm0.25}$ | $6.70_{\pm3.45}$ | $6.82_{\pm2.64}$ | $\mathbf{6.12}_{\pm0.78}$ |

Table 4: Average calibration performance under cross-dataset and domain-generalization settings.

| Method | Cross-Dataset | | | | Domain-Generalization | | | |
|---|---|---|---|---|---|---|---|---|
| | ECE (Conf) | ECE (CAC) | ACE (Conf) | ACE (CAC) | ECE (Conf) | ECE (CAC) | ACE (Conf) | ACE (CAC) |
| CoCoOp | $4.28_{\pm0.23}$ | $\mathbf{4.15}_{\pm0.12}$ | $4.39_{\pm0.43}$ | $\mathbf{4.15}_{\pm0.53}$ | $3.97_{\pm0.19}$ | $\mathbf{3.37}_{\pm0.23}$ | $3.96_{\pm0.51}$ | $\mathbf{3.34}_{\pm0.57}$ |
| KgCoOp | $4.37_{\pm0.04}$ | $\mathbf{4.15}_{\pm0.42}$ | $4.33_{\pm0.33}$ | $\mathbf{4.11}_{\pm0.04}$ | $4.49_{\pm0.45}$ | $\mathbf{2.96}_{\pm0.37}$ | $4.53_{\pm0.09}$ | $\mathbf{3.00}_{\pm0.09}$ |
| MaPLe | $4.04_{\pm0.04}$ | $\mathbf{3.72}_{\pm0.32}$ | $4.07_{\pm0.05}$ | $\mathbf{3.76}_{\pm0.23}$ | $4.10_{\pm0.03}$ | $\mathbf{2.94}_{\pm0.19}$ | $4.18_{\pm0.39}$ | $\mathbf{3.01}_{\pm0.02}$ |
| ProGrad | $4.16_{\pm0.02}$ | $\mathbf{4.03}_{\pm0.07}$ | $4.06_{\pm0.49}$ | $\mathbf{4.04}_{\pm0.03}$ | $4.44_{\pm0.26}$ | $\mathbf{3.18}_{\pm0.10}$ | $4.39_{\pm0.19}$ | $\mathbf{3.17}_{\pm0.47}$ |
| PromptSRC | $4.40_{\pm0.06}$ | $\mathbf{3.92}_{\pm0.40}$ | $4.33_{\pm0.25}$ | $\mathbf{3.93}_{\pm0.06}$ | $4.10_{\pm0.23}$ | $\mathbf{3.89}_{\pm0.02}$ | $4.09_{\pm0.12}$ | $\mathbf{3.94}_{\pm0.02}$ |

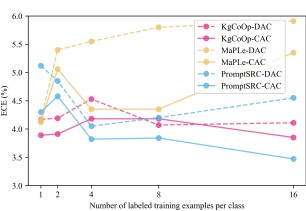

Figure 2: Calibration performance using different shots.

Table 6: Average calibration performance across 11 datasets on different backbones.

| Method | RN50 | | ViT-B16 | | ViT-B32 | |
|---|---|---|---|---|---|---|
| | DAC | CAC | DAC | CAC | DAC | CAC |
| KgCoOp | 4.11 | **3.96** | 4.11 | **3.85** | 8.07 | **4.59** |
| MaPLe | – | – | 5.91 | **5.35** | 4.22 | **4.01** |
| PromptSRC | 9.70 | **7.87** | 3.55 | **3.47** | 5.38 | 5.27 |

our method, CAC, aligns the visual-language representations of the fine-tuned and original models, and it is more adaptable compared to DAC. The results presented in Table 4 indicate our method provides unified performance improvements for all methods, demonstrating the robustness of CAC. In Summary, our method effectively addresses the significant class distribution shifts that traditional calibration methods and DAC struggle to handle, showcasing its adaptability to a wide range of tasks and diverse settings. Detailed experimental results can be found in the Appendix E.

**RQ3**: Does the CAC successfully improve the calibration performance, as designed?

**The Effect of Each Component in CAC.** We evaluate the effect of each component in CAC on calibration performance by systematically ablating the designed modules and assessing their performance on test datasets as follows:

- **Without $\alpha$**: When $\alpha$ is removed, CAC struggles to handle underconfident datasets, causing poor performance. Therefore, the inclusion of $\alpha$, which expands the range of values, is essential.
- **Without $k$**: Ablating $k$ hinders the models ability to effectively distinguish the contrast differences, resulting in worse calibration, which aligns with our initial design rationale. Therefore, using amplification based on $k$ remains the optimal choice.
- **With other function**: We conducted comparative experiments under the same settings by replacing the EXP function with $1/x$ and $-\ln(x)$ function. Since the values of these functions fluctuate drastically with respect to $x$, they fail to handle subtle confidence imbalances and may even amplify miscalibration, which highlights the rationale for using the exponential function.
- **Without the advanced module**: As the advanced module is specifically designed for datasets that require substantial confidence calibration, removing it results in the performance of the original version of CAC, which consistently outperforms DAC. Moreover, all fine-tuning methods benefit from the more robust confidence calibration provided by CAC.

**The Effect of Different Shots and Backbones.** To verify the robustness of CACs hyperparameters, we evaluated its performance under varying training variables (shots and backbones) in the open-vocabulary setting using default parameters. As visualized in Figure 2, the ECE performance under various shot settings demonstrates that CAC consistently outperforms DAC, even using default pa-

Table 5: Average calibration performance across 11 datasets in 5 prompt learning methods. "w/" and "w/o" indicate the results of CAC with and without the inclusion of a specific component.

| Method | Conf | DAC | w/o $k$ | w/o $\alpha$ | w/ 1/x | w/ -ln(x) | w/o advanced | CAC |
|---|---|---|---|---|---|---|---|---|
| CoCoOp | 5.44 | 5.70 | 9.08 | 9.73 | 11.15 | 15.68 | 4.72 | **4.24** |
| KgCoOp | 3.98 | 4.11 | 7.52 | 8.12 | 26.74 | 27.89 | 3.97 | **3.85** |
| MaPLe | 7.80 | 5.91 | 11.32 | 11.25 | 28.13 | 21.76 | 6.57 | **5.35** |
| ProGrad | 5.04 | 6.13 | 8.92 | 11.84 | 28.85 | 24.34 | 4.44 | **4.04** |
| PromptSRC | 4.29 | 4.55 | 7.80 | 9.41 | 23.87 | 19.56 | 3.84 | **3.47** |

Table 7: Time overhead analysis for CAC on Caltech101 and ImageNet datasets.

| | Sample size | Test time | | Calibration | Total time | Time per sample | Incremental time per sample | Train time |
|---|---|---|---|---|---|---|---|---|
| | | PromptSRC | CLIP | CAC | | | | |
| Caltech101 | 916 | 1.78s | 1.65s | 0.06s | 3.49s | 0.004s | 0.002s | 7m 55s |
| ImageNet | 25000 | 27.41s | 19.85s | 0.16s | 47.42s | 0.002s | 0.0008s | 4h 30m |

rameters. Moreover, Table 6 further supports this, showing that CAC achieves superior calibration without the need for extensive hyperparameter tuning across different backbone architectures.

**Running Time.** In Table 7, we analyze the time overhead introduced by CAC. Compared to the time-consuming training process training process, the time overhead of CAC during test-time calibration is negligible and further diminishes as the number of samples increases, as observed in datasets such as Caltech101 and ImageNet.

**Reliability of CAC.** To examine the ability of CAC to handle cases where the model logits deviate significantly from the original logits, we perform confidence calibration on a LoRA-fine-tuned CLIP (Zanella & Ben Ayed, 2024). As shown in Table 8, CAC demonstrates strong calibration performance (e.g., the ECE on DTD decreases from 23.40 to 5.56), indicating that the well-calibrated logits of the original CLIP provide a solid foundation for CAC.

**Parameter Selection Guidelines.** To address more challenging real-world scenarios, we provide a detailed analysis of the optimal parameter choices for each module in **Appendix D**. Based on this analysis, we summarize the following parameter selection guidelines for CAC: for all baselines and datasets in the open-vocabulary setting, the default parameters ($\alpha = 1.10, k = 15$) are generally sufficient, reflecting the parameter insensitivity of CAC. For different baselines or evaluation settings, $\alpha$ can be adjusted based on reliability diagramsspecifically, increasing $\alpha$ for underconfident models and decreasing it for overconfident ones, while $k$ primarily scales the input magnitude. Overall, CAC effectively integrates our research findings into a unified confidence calibration framework, with extensive experiments confirming its parameter robustness and adaptability.

## 6 CONCLUSION

Prompt learning for vision-language models (VLMs) has gained significant attention; however, fine-tuned VLMs face substantial calibration challenges in open-vocabulary settings. Existing calibration methods struggle to address unseen classes, while recently proposed VLM-specific calibration approaches encounter inability to handle train classes and suboptimal performance. In this paper, we present a comprehensive study on VLM calibration in open-vocabulary settings. Through empirical analysis, we identify poor intra-class and inter-class discriminative ability on unseen classes as the primary cause of miscalibration. To address this, we establish a connection between contrast metric and confidence calibration and then propose the Contrast-Aware Calibration (CAC) method. Extensive experimental results demonstrate that our proposal consistently achieves SOTA calibration performance on both train and unseen classes, without compromising accuracy. Moreover, CAC demonstrated parameter insensitivity and robustness in various settings.

As analyzed in our experiments, a potential limitation of our method is that it requires an additional forward pass through CLIP to compute the logits. In future work, we plan to investigate the performance of CAC on other VLMs and further explore more efficient calibration approaches.

ETHICS STATEMENT

This work aims to enhance the confidence calibration performance of fine-tuned VLMs. It does not involve human subjects, sensitive personal data. All datasets used are publicly available, and we adhere to the ICLR Code of Ethics.

REPRODUCIBILITY STATEMENT

We have taken extensive measures to ensure the reproducibility of our results. An anonymous code repository is available at `https://anonymous.4open.science/r/Calibration-E758/`, containing implementations of our calibration methods and running scripts. Details of experimental settings are described in Subsection 5.1 and Appendix B. All datasets used are publicly available. Additional ablation studies and extended experimental results are included in Appendix D and Appendix E.

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

## A   USE OF LARGE LANGUAGE MODELS (LLMS)

In this work, LLMs were used solely for language polishing and translation; they did not contribute to research ideation, experiment design, or result interpretation.

## B   EXPERIMENTAL SETTING

**Dataset.** Following the CoCoOp framework (Zhou et al., 2022a), we conducted evaluations of our proposed CAC along with comparison calibration methods on various image classification tasks. These tasks included general object recognition using ImageNet (Deng et al., 2009) and Caltech101 (Fei-Fei et al., 2004) datasets, fine-grained object recognition involving datasets such as Oxford Pets (Parkhi et al., 2012), Food-101 (Bossard et al., 2014), Stanford Cars (Krause et al., 2013), Oxford Flowers102 (Nilsback & Zisserman, 2008), and FGVCAircraft (Maji et al., 2023). Additionally, we performed a remote sensing recognition task using the EuroSAT (Helber et al., 2019) dataset, a texture recognition task using the DTD (Cimpoi et al., 2014) dataset, an action recognition task using UCF101 (Soomro et al., 2012) dataset and a large-scale scene understanding task using SUN397 (Xiao et al., 2010) dataset.

**Implementation details.** In our experiments, we utilize CLIP (ViT-B/16) as the pre-trained VLM and report results averaged over 3 runs. The model is fine-tuned in a few-shot setting (Zhou et al., 2022b) with 16 samples per class. The general hyperparameters are listed in Table 9, as per the official implementation. Below, we outline the specific hyperparameters for each VLM tuning method. All methods are adopted from their official implementations. For CoCoOp, no additional hyperparameters are required. In the case of ProDA, we set $\lambda = 0.1$. For KgCoOp, $\lambda$ is set to 8.0. MaPLe is configured with a prompt depth $J = 0$, and both the language and vision prompt lengths are set to 2. For ProGrad, $\lambda = 0.8$ is used. For PromptSRC, deep prompting is set with $V = T = 4$, and weight loss employs $\lambda_1 = 10$ and $\lambda_2 = 25$. Finally, for textual diversity, we use $N = 60$ standard prompt templates. All methods are reproduced using the standard codebase. All experiments are conducted on a single NVIDIA A800 GPU according to the standard codebase requirements

## C   DETAILED ANALYSIS ABOUT OBSERVATIONS

We extract features using CLIP (ViT-B/16) and visualize them with t-SNE (Figure 3), using ImageNet as a reference to compare feature distributions across datasets. It can be observed that the features of different categories in FGVCAircraft are heavily intertwined, indicating high inter-class similarity. When fine-tuned VLMs are applied for classification, they tend to assign most samples to a few dominant categories, leading to underconfidence in the remaining categories. Numerically, this is reflected as $s^+ < s^-$, i.e., a negative contrast score, which indicates a severe confidence imbalance. In contrast, for Food101, the features exhibit a clustering tendency for each class, but with larger intra-class variance compared to ImageNet, reflecting high intra-class variation and low inter-class similarity. When fine-tuned VLMs are used for classification, multiple categories may receive high scores simultaneously, but the correct class can still be identified. This corresponds to $s^+ > s^-$, i.e., a positive contrast score, though with a relatively small margin.

## D   DETAILED ABLATION STUDIES

### D.1   THE IMPACT OF DIFFERENT $k$

Intuitively, scaling the metric $z$ to an appropriate range produces optimal confidence calibration results, while excessively high or low values lead to suboptimal performance. Table 10 confirms our hypothesis, showing that a proper value of $k$ achieves the best calibration performance. In general, this experiment highlights that the regularized $L_1$ distance tends to be relatively small, requiring appropriate amplification for optimal results.

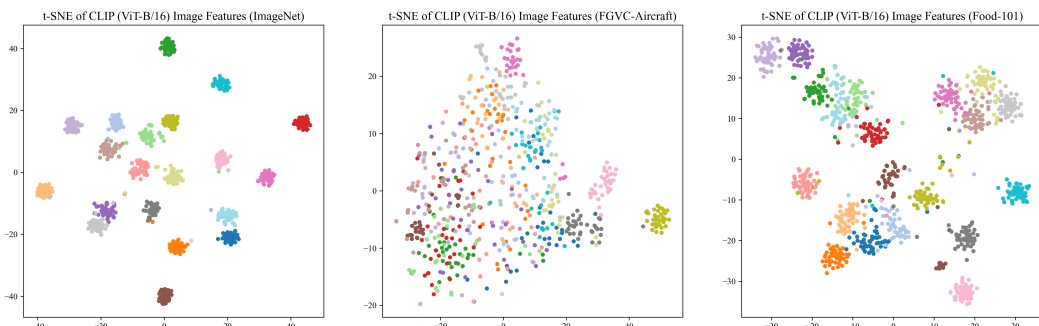

Figure 3: t-SNE visualization of CLIP (ViT-B/16) features across ImageNet, FGVCAircraft and Food101 datasets.

### D.2 THE IMPACT OF DIFFERENT $\alpha$ VALUES

As shown in Table 11, we tested the effect of 4 different $\alpha$ values, with the default setting of $\alpha$ being 1.10. When $\alpha = 1.00$, the model cannot handle underconfidence, resulting in degraded calibration performance. As $\alpha$ increases, CACs ability to mitigate underconfidence improves. However, when $\alpha \geq 1.20$, the cases of underconfidence are largely neglected, leading to CAC calibration to change to overconfidence, which compromises its balance between underconfidence and overconfidence and leads to suboptimal results. In summary, as intended in our design, $\alpha$ plays a critical role in the models ability to address both underconfidence and overconfidence.

### D.3 IMPACT OF PIECEWISE FUNCTION THRESHOLDS ON CONFIDENCE CALIBRATION

For the overconfidence threshold $\lambda_1$, we set 0.90 as the default scaling coefficient. As shown in Table 12, values below $\lambda_1$ indicate a significant deviation between the model output and the original CLIP. For the underconfidence threshold $\lambda_2$, we hypothesize that when the scores of a fine-tuned CLIP closely align with those of the original CLIP, its predictions are more reliable; therefore, we select 1.00 as the underconfidence threshold. We also tested two alternative thresholds, 0.95 and 1.05, which showed minimal differences in results, suggesting that most outputs resembling the original CLIP's predictions are underconfident. In summary, the minimal impact of the piecewise function thresholds highlights that CAC weights are the key factor in achieving excellent confidence calibration, further validating the effectiveness.

## E  DETAILED EXPERIMENTAL RESULTS

In this section, we provide a comprehensive details of the experimental results, with a focus on the calibration and performance of the fine-tuned VLMs. First, we evaluate the calibration performance on both train and unseen classes, comparing the fine-tuning results. Next, we explore the calibration performance, discussing how advanced calibration techniques enhance the models reliability in making accurate predictions. Additionally, we delve into the relationship between accuracy and confidence. Finally, we present detailed results from open-vocabulary se, showcasing the models performance across various tasks and highlighting the improvements achieved through CAC.

### E.1 THE CALIBRATION PERFORMANCE OF FINE-TUNED CLIP

To illustrate the miscalibration in fine-tuned VLMs, we fine-tune the pre-trained CLIP using 5 different tuning methods across 11 downstream datasets, evaluating calibration performance with ECE. In this section, the detailed calibration results are provided. As can be seen from Table 13, it is evident that distillation-based methods (KgCoOp, ProGrad, and PromptSRC) play a role in restricting the model's confidence regarding train classes. For unseen classes, distillation-based methods leverage zero-shot CLIP as a teacher model to enhance generalization, leading to more reliable predictions for these unseen classes, as expected. Through comparison, we find that fine-tuned VLMs tend to

be underconfident on the train classes, while they are often overconfident on unseen classes. This observation motivates a deeper investigation into the calibration of fine-tuned VLMs.

## E.2    CALIBRATED PERFORMANCE FOR FINE-TUNED CLIP

In our study, we apply commonly employed post-hoc confidence calibration techniques to fine-tuned VLMs, aiming to calibrate both train and unseen classes. Our findings reveal that the established calibration methods are capable of rectifying the miscalibration issues within the train classes. Nevertheless, this effectiveness fails to extend to the unseen classes.

### E.2.1    TRAIN CLASS

**Post-hoc calibration can remedy miscalibration in train classes.** Recent studies (Zadrozny & Elkan, 2001; 2002) have successfully extended scaling-based methods, such as Temperature Scaling (TS), to the calibration of VLMs, demonstrating their effectiveness in addressing miscalibration in near-world environments. Additionally, we observe that sample-based calibration methods, like MIR, significantly reduce miscalibration for train classes. However, as shown in Table 15, most traditional methods exhibit unstable behavior in VLM confidence calibration tasks, lacking consistency across different fine-tuning strategies and datasets. In contrast, CAC, specifically designed for VLMs, not only achieves optimal calibration for unseen classes but also generalizes well to train classes, offering a more robust and reliable solution.

**Post-hoc calibration for train classes does not extend to unseen classes.** For instance, scaling-based methods like TS adjust logits by a single scalar temperature value. When a fine-tuned VLM exhibits underconfidence in the train classes, TS sharpens the logits before the softmax function, increasing confidence in predictions. However, this comes at the cost of making the confidence distribution more peaked, which leads to overconfidence in unseen classes. Bin-based methods, on the other hand, require probabilities from train classes as input, which is incompatible with zero-probability predictions. This validates our motivation and highlights the rationale behind leveraging zero-shot CLIP in our approach.

### E.2.2    UNSEEN CLASS

Detailed experimental results can be found in Table 14.

## E.3    ACCURACY & CONFIDENCE OF FINE-TUNED CLIP

This section is dedicated to the analysis of miscalibration in the context of classification performance. Since open-vocabulary classes are not encountered during tuning, some might hypothesize that miscalibration stems from a sharp accuracy drop, leading to a higher ECE. We report the average performance across 11 datasets in Table 16, and provide a comparison of confidence calibration in Table 17. While classification performance on unseen classes may be comparable or lower, fine-tuned VLMs still have higher average predictive confidence than zero-shot CLIP, resulting in a larger ECE. This aligns with CAC results: although fine-tuning boosts accuracy, it also increases ECE.

## E.4    DETAILED RESULTS OF OPEN-VOCABULARY SETTING

This section showcases detailed results of calibration for unseen classes, aiming to illustrate that CAC facilitates open-vocabulary calibration in current prompt tuning. For a comprehensive evaluation, we use four standard metrics to assess open-vocabulary confidence calibration: Expected Calibration Error (ECE) (Guo et al., 2017a), Maximum Calibration Error (MCE) (Guo et al., 2017a), Adaptive Calibration Error (ACE) (Nixon et al., 2019), and Proximity Informed Expected Calibration Error (PIECE) (Xiong et al., 2023). Results can be found in Tables 18, 19, 20 and 21.

Table 8: The ECE results of LoRA fine-tuning CLIP by CAC.

| Method | Caltech101 | DTD | EuroSAT | FGVC | Food101 | ImageNet | OF | OP | CARS | UCF | SUN | AVG |
|---|---|---|---|---|---|---|---|---|---|---|---|---|
| CLIP | 2.21 | 8.40 | 6.92 | 5.22 | 2.50 | 1.94 | 3.14 | 7.10 | 4.07 | 1.98 | 2.81 | 4.21 |
| LoRA(Conf) | 2.49 | 23.40 | 11.81 | 16.43 | 9.38 | 3.48 | 4.09 | 5.76 | 2.79 | 7.85 | 7.13 | 8.60 |
| LoRA(CAC) | 1.49 | 5.56 | 3.26 | 6.50 | 2.92 | 2.53 | 2.41 | 5.53 | 3.12 | 5.50 | 2.38 | 3.75 |

Table 9: Hyper-parameter settings for different models. All of fine-tuning method is rigorously reproduced based on the experimental details given in its paper.

| Method | Prompt Depth | Prompt Length | Epochs | Warmup Epochs | Learning Rate | Batch Size |
|---|---|---|---|---|---|---|
| CoCoOp | 1 | 4 | 20 | 1 | 0.002 | 1 |
| KgCoOp | 1 | 16 | 200 | 1 | 0.002 | 32 |
| MaPLe | 9 | 2 | 5 | 1 | 0.0026 | 4 |
| ProGrad | 1 | 16 | 100 | 1 | 0.002 | 32 |
| PromptSRC | 9 | 4 | 50 | 1 | 0.0025 | 4 |

Table 10: Average calibration performance of different $k$ of CAC across 11 datasets.

| Method | Conf | 10 | 15 | 20 | 25 |
|---|---|---|---|---|---|
| CoCoOp | 5.44 | 4.82 | **4.24** | 6.41 | 11.09 |
| KgCoOp | 3.98 | 3.82 | 3.85 | **3.66** | 5.16 |
| MaPLe | 7.80 | 6.82 | **5.35** | 8.14 | 12.52 |
| ProGrad | 5.04 | 4.57 | **4.04** | 8.20 | 12.74 |
| PromptSRC | 4.29 | 3.93 | **3.47** | 5.33 | 8.31 |

Table 11: Average calibration performance of different $\alpha$ of CAC across 11 datasets.

| Method | Conf | 1.00 | 1.05 | 1.10 | 1.20 |
|---|---|---|---|---|---|
| CoCoOp | 5.44 | 7.87 | 5.88 | **4.24** | 4.84 |
| KgCoOp | 3.98 | 5.86 | **3.81** | 3.85 | 4.73 |
| MaPLe | 7.80 | 9.13 | 6.93 | **5.35** | 5.85 |
| ProGrad | 5.04 | 10.38 | 7.09 | **4.04** | 4.29 |
| PromptSRC | 4.29 | 7.87 | 5.00 | **3.47** | 3.90 |

Table 12: The impact of different piecewise function thresholds on CAC confidence calibration.

| Method | $\lambda_1$ | | $\lambda_2$ | | CAC |
|---|---|---|---|---|---|
| | 0.85 | 0.95 | 0.95 | 1.05 | |
| CoCoOp | 5.63 | 4.84 | 4.63 | 4.57 | 4.24 |
| KgCoOp | 4.22 | 3.55 | 3.72 | 3.59 | 3.85 |
| MaPLe | 7.88 | 5.92 | 5.71 | 5.69 | 5.35 |
| ProGrad | 5.19 | 5.27 | 4.71 | 4.65 | 4.04 |
| PromptSRC | 4.41 | 4.12 | 3.83 | 3.69 | 3.47 |

Table 13: Expected Calibration Error (ECE) on diverse downstream datasets using various tuning methods for CLIP-ViT-B/16. The calibration performance is averaged across three variants.

(a) Train

| Method | Caltech101 | DTD | EuroSAT | FGVC | Food101 | ImageNet | OF | OP | CARS | UCF | SUN | AVG |
|---|---|---|---|---|---|---|---|---|---|---|---|---|
| ZeroshotCLIP | 6.48 | 4.71 | 8.29 | 2.71 | 1.57 | 1.53 | 3.09 | 2.25 | 3.80 | 1.54 | 3.43 | 3.58 |
| CoCoOp | 1.45 | 2.20 | 7.19 | 3.82 | 0.87 | 2.65 | 7.74 | 2.33 | 7.68 | 1.51 | 2.17 | 3.60 |
| KgCoOp | 2.42 | 7.45 | 11.14 | 6.81 | 1.38 | 2.64 | 9.91 | 2.93 | 10.90 | 4.88 | 4.16 | 5.87 |
| MaPLe | 1.60 | 3.47 | 2.77 | 4.27 | 0.73 | 1.89 | 4.07 | 2.16 | 7.20 | 0.95 | 1.72 | 2.80 |
| ProGrad | 2.94 | 11.86 | 9.94 | 6.51 | 1.41 | 2.66 | 9.14 | 2.97 | 9.45 | 3.74 | 4.56 | 5.93 |
| PromptSRC | 2.37 | 3.10 | 9.17 | 3.25 | 0.82 | 2.12 | 4.75 | 2.88 | 8.28 | 2.08 | 2.35 | 3.74 |

(b) Unseen

| Method | Caltech101 | DTD | EuroSAT | FGVC | Food101 | ImageNet | OF | OP | CARS | UCF | SUN | AVG |
|---|---|---|---|---|---|---|---|---|---|---|---|---|
| ZeroshotCLIP | 1.59 | 9.45 | 9.18 | 6.57 | 1.79 | 2.10 | 4.94 | 3.42 | 3.22 | 3.55 | 5.28 | 4.64 |
| CoCoOp | 2.44 | 13.27 | 10.20 | 13.43 | 1.91 | 1.62 | 7.16 | 2.31 | 2.09 | 1.73 | 3.73 | 5.44 |
| KgCoOp | 1.47 | 5.10 | 7.18 | 10.74 | 1.94 | 1.91 | 4.80 | 3.28 | 3.22 | 1.21 | 2.97 | 3.98 |
| MaPLe | 2.45 | 20.77 | 17.18 | 16.52 | 1.16 | 2.99 | 14.67 | 2.59 | 3.01 | 2.10 | 2.41 | 7.80 |
| ProGrad | 2.33 | 5.81 | 16.81 | 10.19 | 1.52 | 2.55 | 5.30 | 2.89 | 3.46 | 1.06 | 3.48 | 5.04 |
| PromptSRC | 1.85 | 4.83 | 7.79 | 15.41 | 1.53 | 1.63 | 5.92 | 2.98 | 1.87 | 0.94 | 2.47 | 4.29 |

Table 14: Expected Calibration Error (ECE) on unseen classes in diverse downstream datasets using various tuning methods for CLIP-ViT-B/16, with calibration performance averaged across three variants.

(a) W/O Calibration

| Method | Caltech101 | DTD | EuroSAT | FGVC | Food101 | ImageNet | OF | OP | CARS | UCF | SUN | AVG |
|---|---|---|---|---|---|---|---|---|---|---|---|---|
| ZeroshotCLIP | 1.59 | 9.45 | 9.18 | 6.57 | 1.79 | 2.10 | 4.94 | 3.42 | 3.22 | 3.55 | 5.28 | 4.64 |
| CoCoOp | 2.44 | 13.27 | 10.20 | 13.43 | 1.91 | 1.62 | 7.16 | 2.31 | 2.09 | 1.73 | 3.73 | 5.44 |
| KgCoOp | 1.47 | 5.10 | 7.18 | 10.74 | 1.94 | 1.91 | 4.80 | 3.28 | 3.22 | 1.21 | 2.97 | 3.98 |
| MaPLe | 2.45 | 20.77 | 17.18 | 16.52 | 1.16 | 2.99 | 14.67 | 2.59 | 3.01 | 2.10 | 2.41 | 7.80 |
| ProGrad | 2.33 | 5.81 | 16.81 | 10.19 | 1.52 | 2.55 | 5.30 | 2.89 | 3.46 | 1.06 | 3.48 | 5.04 |
| PromptSRC | 1.85 | 4.83 | 7.79 | 15.41 | 1.53 | 1.63 | 5.92 | 2.98 | 1.87 | 0.94 | 2.47 | 4.29 |

(b) DAC

| Method | Caltech101 | DTD | EuroSAT | FGVC | Food101 | ImageNet | OF | OP | CARS | UCF | SUN | AVG |
|---|---|---|---|---|---|---|---|---|---|---|---|---|
| ZeroshotCLIP | 1.59 | 9.45 | 9.18 | 6.57 | 1.79 | 2.10 | 4.94 | 3.42 | 3.22 | 3.55 | 5.28 | 4.64 |
| CoCoOp | 2.18 | 6.26 | 9.30 | 3.52 | 3.81 | 7.85 | 5.94 | 3.88 | 6.42 | 8.28 | 5.26 | 5.70 |
| KgCoOp | 1.44 | 4.03 | 9.11 | 9.32 | 2.55 | 1.90 | 4.38 | 3.55 | 3.57 | 1.58 | 3.77 | 4.11 |
| MaPLe | 1.91 | 6.25 | 10.65 | 8.53 | 3.38 | 2.57 | 8.60 | 4.23 | 4.91 | 6.55 | 7.46 | 5.91 |
| ProGrad | 2.88 | 10.67 | 8.44 | 3.03 | 4.24 | 3.12 | 4.42 | 5.28 | 6.04 | 8.42 | 10.93 | 6.13 |
| PromptSRC | 2.10 | 4.53 | 9.59 | 12.48 | 2.58 | 1.59 | 5.59 | 3.55 | 2.51 | 2.81 | 2.71 | 4.55 |

(c) CAC

| Method | Caltech101 | DTD | EuroSAT | FGVC | Food101 | ImageNet | OF | OP | CARS | UCF | SUN | AVG |
|---|---|---|---|---|---|---|---|---|---|---|---|---|
| ZeroshotCLIP | 2.17 | 8.84 | 7.08 | 10.99 | 0.77 | 5.92 | 5.89 | 1.87 | 1.35 | 2.38 | 2.35 | 4.51 |
| CoCoOp | 2.45 | 7.80 | 9.71 | 8.62 | 1.74 | 1.54 | 4.21 | 3.48 | 2.68 | 1.20 | 3.18 | 4.24 |
| KgCoOp | 2.02 | 5.29 | 6.39 | 8.96 | 0.30 | 4.85 | 5.10 | 2.34 | 2.90 | 1.14 | 3.01 | 3.85 |
| MaPLe | 2.26 | 10.31 | 16.22 | 8.63 | 1.21 | 2.30 | 5.57 | 2.95 | 4.52 | 1.96 | 2.94 | 5.35 |
| ProGrad | 2.28 | 5.17 | 10.7 | 5.6 | 1.64 | 2.15 | 3.77 | 3.69 | 3.39 | 2.03 | 4.01 | 4.04 |
| PromptSRC | 1.79 | 4.14 | 7.82 | 7.89 | 0.92 | 2.22 | 5.83 | 2.29 | 1.86 | 1.24 | 2.19 | 3.47 |

Table 15: Expected Calibration Error (ECE) on train classes in diverse downstream datasets using various tuning methods for CLIP-ViT-B/16, with calibration performance averaged across three variants.

(a) W/O Calibration

| Method | Caltech101 | DTD | EuroSAT | FGVC | Food101 | ImageNet | OF | OP | CARS | UCF | SUN | AVG |
|---|---|---|---|---|---|---|---|---|---|---|---|---|
| ZeroshotCLIP | 6.48 | 4.71 | 8.29 | 2.71 | 1.57 | 1.53 | 3.09 | 2.25 | 3.80 | 1.54 | 3.43 | 3.58 |
| CoCoOp | 1.45 | 2.20 | 7.19 | 3.82 | 0.87 | 2.65 | 7.74 | 2.33 | 7.68 | 1.51 | 2.17 | 3.60 |
| KgCoOp | 2.42 | 7.45 | 11.14 | 6.81 | 1.38 | 2.64 | 9.91 | 2.93 | 10.90 | 4.88 | 4.16 | 5.87 |
| MaPLe | 1.60 | 3.47 | 2.77 | 4.27 | 0.73 | 1.89 | 4.07 | 2.16 | 7.20 | 0.95 | 1.72 | 2.80 |
| ProGrad | 2.94 | 11.86 | 9.94 | 6.51 | 1.41 | 2.66 | 9.14 | 2.97 | 9.45 | 3.74 | 4.56 | 5.93 |
| PromptSRC | 2.37 | 3.10 | 9.17 | 3.25 | 0.82 | 2.12 | 4.75 | 2.88 | 8.28 | 2.08 | 2.35 | 3.74 |

(b) Isotonic Regression

| Method | Caltech101 | DTD | EuroSAT | FGVC | Food101 | ImageNet | OF | OP | CARS | UCF | SUN | AVG |
|---|---|---|---|---|---|---|---|---|---|---|---|---|
| ZeroshotCLIP | 3.11 | 7.39 | 10.41 | 8.30 | 6.34 | 2.04 | 4.85 | 6.58 | 9.20 | 7.82 | 11.73 | 7.07 |
| CoCoOp | 4.90 | 5.79 | 8.11 | 15.66 | 10.55 | 1.48 | 6.88 | 6.68 | 9.16 | 7.65 | 8.92 | 7.80 |
| KgCoOp | 1.29 | 4.80 | 11.50 | 11.93 | 9.81 | 0.99 | 18.49 | 8.78 | 3.80 | 4.57 | 3.54 | 7.23 |
| MaPLe | 4.00 | 7.60 | 6.87 | 15.29 | 12.39 | 1.47 | 2.73 | 7.69 | 8.85 | 7.95 | 11.06 | 7.81 |
| ProGrad | 2.90 | 3.78 | 6.61 | 13.41 | 8.10 | 1.19 | 5.71 | 3.64 | 4.84 | 5.39 | 7.05 | 5.69 |
| PromptSRC | 2.28 | 3.78 | 5.98 | 14.33 | 10.78 | 1.41 | 4.89 | 4.62 | 5.12 | 6.91 | 10.22 | 6.39 |

(c) Histogram Binning

| Method | Caltech101 | DTD | EuroSAT | FGVC | Food101 | ImageNet | OF | OP | CARS | UCF | SUN | AVG |
|---|---|---|---|---|---|---|---|---|---|---|---|---|
| ZeroshotCLIP | 2.16 | 11.40 | 9.76 | 2.23 | 2.87 | 4.73 | 8.24 | 1.61 | 8.65 | 5.03 | 7.81 | 5.86 |
| CoCoOp | 3.18 | 17.48 | 7.37 | 4.04 | 2.76 | 4.68 | 17.03 | 5.93 | 5.98 | 6.21 | 7.89 | 7.50 |
| KgCoOp | 0.76 | 13.78 | 14.73 | 4.80 | 2.38 | 4.37 | 15.67 | 5.16 | 5.52 | 5.37 | 9.00 | 7.41 |
| MaPLe | 0.94 | 12.45 | 13.01 | 7.28 | 2.90 | 4.28 | 14.31 | 1.32 | 7.25 | 3.68 | 5.46 | 6.63 |
| ProGrad | 1.08 | 11.66 | 4.20 | 5.05 | 2.41 | 4.06 | 13.51 | 1.94 | 6.46 | 5.35 | 6.22 | 5.63 |
| PromptSRC | 1.05 | 14.46 | 13.01 | 4.02 | 2.37 | 4.11 | 16.20 | 1.60 | 4.69 | 3.97 | 4.70 | 6.38 |

(d) Multi-Isotonic Regression

| Method | Caltech101 | DTD | EuroSAT | FGVC | Food101 | ImageNet | OF | OP | CARS | UCF | SUN | AVG |
|---|---|---|---|---|---|---|---|---|---|---|---|---|
| ZeroshotCLIP | 3.47 | 3.68 | 2.84 | 1.97 | 2.80 | 0.31 | 2.90 | 2.56 | 1.78 | 1.93 | 4.45 | 2.61 |
| CoCoOp | 0.38 | 5.07 | 4.09 | 4.80 | 2.06 | 1.43 | 10.33 | 1.83 | 6.97 | 2.42 | 3.24 | 3.87 |
| KgCoOp | 0.74 | 12.33 | 19.93 | 5.27 | 1.27 | 2.11 | 16.35 | 3.76 | 8.30 | 5.31 | 5.80 | 7.38 |
| MaPLe | 0.35 | 4.02 | 3.12 | 4.68 | 2.87 | 0.98 | 4.24 | 1.62 | 3.82 | 2.13 | 1.90 | 2.70 |
| ProGrad | 0.79 | 9.26 | 5.19 | 4.94 | 1.46 | 0.93 | 11.75 | 2.02 | 5.41 | 2.95 | 3.89 | 4.42 |
| PromptSRC | 0.60 | 4.14 | 10.26 | 4.11 | 1.57 | 0.69 | 7.04 | 1.14 | 6.07 | 1.60 | 1.88 | 3.55 |

(e) CAC

| Method | Caltech101 | DTD | EuroSAT | FGVC | Food101 | ImageNet | OF | OP | CARS | UCF | SUN | AVG |
|---|---|---|---|---|---|---|---|---|---|---|---|---|
| ZeroshotCLIP | 4.93 | 4.85 | 9.11 | 2.76 | 0.43 | 2.30 | 4.69 | 1.19 | 1.38 | 3.57 | 4.05 | 3.57 |
| CoCoOp | 1.10 | 3.44 | 5.83 | 2.97 | 0.46 | 3.29 | 6.22 | 1.61 | 5.61 | 1.09 | 1.94 | 3.05 |
| KgCoOp | 1.70 | 4.79 | 9.16 | 4.67 | 0.50 | 1.12 | 7.91 | 1.94 | 8.49 | 2.21 | 2.60 | 4.10 |
| MaPLe | 1.26 | 4.78 | 2.29 | 3.96 | 0.64 | 1.45 | 3.04 | 1.53 | 5.24 | 1.76 | 1.58 | 2.50 |
| ProGrad | 2.30 | 8.81 | 7.97 | 4.26 | 0.46 | 1.09 | 7.19 | 2.15 | 7.31 | 1.40 | 2.89 | 4.17 |
| PromptSRC | 1.69 | 2.77 | 7.25 | 2.94 | 0.61 | 0.94 | 3.66 | 1.63 | 6.42 | 0.71 | 1.58 | 2.75 |

Table 16: Accuracy comparison of existing prompt tuning in the open-vocabulary setting.

(a) Train

| Method | Caltech101 | DTD | EuroSAT | FGVC | Food101 | ImageNet | OF | OP | CARS | UCF | SUN | AVG |
|---|---|---|---|---|---|---|---|---|---|---|---|---|
| ZeroshotCLIP | 97.16 | 53.24 | 57.00 | 27.61 | 90.06 | 72.42 | 71.70 | 91.33 | 63.77 | 69.32 | 70.89 | 69.50 |
| CoCoOp | 97.91 | 77.16 | 85.61 | 35.07 | 90.53 | 75.89 | 94.24 | 95.04 | 71.67 | 79.29 | 81.94 | 80.40 |
| KgCoOp | 97.89 | 80.71 | 89.51 | 39.36 | 90.61 | 75.98 | 96.20 | 94.95 | 75.07 | 80.95 | 84.33 | 82.32 |
| MaPLe | 98.00 | 80.40 | 89.25 | 33.29 | 90.48 | 76.57 | 96.39 | 95.66 | 72.87 | 81.01 | 84.01 | 81.63 |
| ProGrad | 98.39 | 76.43 | 90.02 | 41.16 | 90.38 | 76.96 | 96.17 | 94.95 | 78.52 | 81.18 | 84.85 | 82.64 |
| PromptSRC | 98.39 | 83.49 | 92.96 | 44.36 | 90.62 | 78.00 | 98.10 | 95.48 | 80.65 | 83.02 | 87.63 | 84.79 |

(b) Unseen

| Method | Caltech101 | DTD | EuroSAT | FGVC | Food101 | ImageNet | OF | OP | CARS | UCF | SUN | AVG |
|---|---|---|---|---|---|---|---|---|---|---|---|---|
| ZeroshotCLIP | 94.14 | 60.87 | 64.00 | 35.93 | 91.15 | 68.11 | 77.50 | 97.15 | 74.94 | 75.59 | 78.42 | 74.35 |
| CoCoOp | 93.52 | 55.35 | 63.04 | 32.81 | 91.53 | 70.41 | 72.17 | 97.44 | 73.50 | 76.61 | 73.77 | 72.74 |
| KgCoOp | 94.36 | 51.29 | 69.22 | 30.59 | 91.57 | 69.58 | 73.21 | 97.45 | 74.12 | 75.18 | 74.09 | 72.79 |
| MaPLe | 93.92 | 46.26 | 52.13 | 22.32 | 91.85 | 69.67 | 67.50 | 97.73 | 73.49 | 77.85 | 78.73 | 70.13 |
| ProGrad | 93.60 | 52.05 | 55.10 | 31.45 | 89.40 | 67.06 | 73.43 | 97.07 | 69.16 | 72.29 | 70.85 | 70.13 |
| PromptSRC | 94.07 | 62.88 | 74.13 | 27.99 | 91.37 | 70.33 | 77.23 | 97.18 | 75.03 | 78.96 | 78.83 | 75.27 |

Table 17: Confidence comparison of existing prompt tuning in the open-vocabulary setting.

(a) Train

| Method | Caltech101 | DTD | EuroSAT | FGVC | Food101 | ImageNet | OF | OP | CARS | UCF | SUN | AVG |
|---|---|---|---|---|---|---|---|---|---|---|---|---|
| ZeroshotCLIP | 0.92 | 0.56 | 0.66 | 0.29 | 0.90 | 0.74 | 0.76 | 0.90 | 0.63 | 0.73 | 0.75 | 0.71 |
| CoCoOp | 0.97 | 0.81 | 0.82 | 0.35 | 0.92 | 0.78 | 0.90 | 0.95 | 0.68 | 0.82 | 0.84 | 0.80 |
| KgCoOp | 0.97 | 0.78 | 0.82 | 0.37 | 0.92 | 0.78 | 0.90 | 0.94 | 0.69 | 0.81 | 0.84 | 0.80 |
| MaPLe | 0.97 | 0.86 | 0.90 | 0.34 | 0.92 | 0.79 | 0.94 | 0.95 | 0.69 | 0.84 | 0.86 | 0.82 |
| ProGrad | 0.97 | 0.71 | 0.84 | 0.39 | 0.92 | 0.79 | 0.91 | 0.94 | 0.73 | 0.82 | 0.84 | 0.81 |
| PromptSRC | 0.97 | 0.84 | 0.86 | 0.43 | 0.91 | 0.78 | 0.95 | 0.94 | 0.74 | 0.83 | 0.87 | 0.83 |

(b) Unseen

| Method | Caltech101 | DTD | EuroSAT | FGVC | Food101 | ImageNet | OF | OP | CARS | UCF | SUN | AVG |
|---|---|---|---|---|---|---|---|---|---|---|---|---|
| ZeroshotCLIP | 0.96 | 0.53 | 0.66 | 0.47 | 0.92 | 0.74 | 0.83 | 0.95 | 0.76 | 0.78 | 0.78 | 0.76 |
| CoCoOp | 0.95 | 0.63 | 0.67 | 0.41 | 0.90 | 0.71 | 0.75 | 0.94 | 0.72 | 0.76 | 0.73 | 0.74 |
| KgCoOp | 0.96 | 0.56 | 0.69 | 0.40 | 0.92 | 0.74 | 0.78 | 0.95 | 0.72 | 0.75 | 0.75 | 0.75 |
| MaPLe | 0.95 | 0.60 | 0.67 | 0.33 | 0.91 | 0.71 | 0.75 | 0.95 | 0.72 | 0.78 | 0.78 | 0.74 |
| ProGrad | 0.95 | 0.51 | 0.66 | 0.38 | 0.88 | 0.68 | 0.77 | 0.94 | 0.69 | 0.71 | 0.70 | 0.72 |
| PromptSRC | 0.95 | 0.65 | 0.69 | 0.35 | 0.91 | 0.72 | 0.83 | 0.95 | 0.74 | 0.78 | 0.79 | 0.76 |

Table 18: Expected Calibration Error (ECE) on unseen classes in diverse downstream datasets using various tuning methods for CLIP-ViT-B/16.

(a) W/O Calibration

| Method | Caltech101 | DTD | EuroSAT | FGVC | Food101 | ImageNet | OF | OP | CARS | UCF | SUN | AVG |
|---|---|---|---|---|---|---|---|---|---|---|---|---|
| ZeroshotCLIP | 1.59 | 9.45 | 9.18 | 6.57 | 1.79 | 2.10 | 4.94 | 3.42 | 3.22 | 3.55 | 5.28 | 4.64 |
| CoCoOp | 2.44 | 13.27 | 10.20 | 13.43 | 1.91 | 1.62 | 7.16 | 2.31 | 2.09 | 1.73 | 3.73 | 5.44 |
| KgCoOp | 1.47 | 5.10 | 7.18 | 10.74 | 1.94 | 1.91 | 4.80 | 3.28 | 3.22 | 1.21 | 2.97 | 3.98 |
| MaPLe | 2.45 | 20.77 | 17.18 | 16.52 | 1.16 | 2.99 | 14.67 | 2.59 | 3.01 | 2.10 | 2.41 | 7.80 |
| ProGrad | 2.33 | 5.81 | 16.81 | 10.19 | 1.52 | 2.55 | 5.30 | 2.89 | 3.46 | 1.06 | 3.48 | 5.04 |
| PromptSRC | 1.85 | 4.83 | 7.79 | 15.41 | 1.53 | 1.63 | 5.92 | 2.98 | 1.87 | 0.94 | 2.47 | 4.29 |

(b) DAC

| Method | Caltech101 | DTD | EuroSAT | FGVC | Food101 | ImageNet | OF | OP | CARS | UCF | SUN | AVG |
|---|---|---|---|---|---|---|---|---|---|---|---|---|
| ZeroshotCLIP | 1.59 | 9.45 | 9.18 | 6.57 | 1.79 | 2.10 | 4.94 | 3.42 | 3.22 | 3.55 | 5.28 | 4.64 |
| CoCoOp | 2.18 | 6.26 | 9.30 | 3.52 | 3.81 | 7.85 | 5.94 | 3.88 | 6.42 | 8.28 | 5.26 | 5.70 |
| KgCoOp | 1.44 | 4.03 | 9.11 | 9.32 | 2.55 | 1.90 | 4.38 | 3.55 | 3.57 | 1.58 | 3.77 | 4.11 |
| MaPLe | 1.91 | 6.25 | 10.65 | 8.53 | 3.38 | 2.57 | 8.60 | 4.23 | 4.91 | 6.55 | 7.46 | 5.91 |
| ProGrad | 2.88 | 10.67 | 8.44 | 3.03 | 4.24 | 3.12 | 4.42 | 5.28 | 6.04 | 8.42 | 10.93 | 6.13 |
| PromptSRC | 2.10 | 4.53 | 9.59 | 12.48 | 2.58 | 1.59 | 5.59 | 3.55 | 2.51 | 2.81 | 2.71 | 4.55 |

(c) CAC

| Method | Caltech101 | DTD | EuroSAT | FGVC | Food101 | ImageNet | OF | OP | CARS | UCF | SUN | AVG |
|---|---|---|---|---|---|---|---|---|---|---|---|---|
| ZeroshotCLIP | 2.17 | 8.84 | 7.08 | 10.99 | 0.77 | 5.92 | 5.89 | 1.87 | 1.35 | 2.38 | 2.35 | 4.51 |
| CoCoOp | 2.45 | 7.80 | 9.71 | 8.62 | 1.74 | 1.54 | 4.21 | 3.48 | 2.68 | 1.20 | 3.18 | 4.24 |
| KgCoOp | 2.02 | 5.29 | 6.39 | 8.96 | 0.30 | 4.85 | 5.10 | 2.34 | 2.90 | 1.14 | 3.01 | 3.85 |
| MaPLe | 2.26 | 10.31 | 16.22 | 8.63 | 1.21 | 2.30 | 5.57 | 2.95 | 4.52 | 1.96 | 2.94 | 5.35 |
| ProGrad | 2.28 | 5.17 | 10.70 | 5.60 | 1.64 | 2.15 | 3.77 | 3.69 | 3.39 | 2.03 | 4.01 | 4.04 |
| PromptSRC | 1.79 | 4.14 | 7.82 | 7.89 | 0.92 | 2.22 | 5.83 | 2.29 | 1.86 | 1.24 | 2.19 | 3.47 |

Table 19: Adaptive Calibration Error (ACE) on unseen classes in diverse downstream datasets using various tuning methods for CLIP-ViT-B/16.

(a) W/O Calibration

| Method | Caltech101 | DTD | EuroSAT | FGVC | Food101 | ImageNet | OF | OP | CARS | UCF | SUN | AVG |
|---|---|---|---|---|---|---|---|---|---|---|---|---|
| ZeroshotCLIP | 1.56 | 9.17 | 9.27 | 6.02 | 1.79 | 2.12 | 4.78 | 3.41 | 3.38 | 3.68 | 5.57 | 4.61 |
| CoCoOp | 2.04 | 13.10 | 10.13 | 13.36 | 1.78 | 1.65 | 7.18 | 2.17 | 1.90 | 1.69 | 3.86 | 5.35 |
| KgCoOp | 1.21 | 4.97 | 7.02 | 10.64 | 1.95 | 1.93 | 5.14 | 3.21 | 3.12 | 1.32 | 2.72 | 3.93 |
| MaPLe | 2.12 | 20.77 | 17.39 | 16.49 | 1.14 | 2.98 | 14.70 | 2.43 | 2.98 | 2.01 | 2.47 | 7.77 |
| ProGrad | 1.95 | 5.54 | 16.73 | 10.14 | 1.41 | 2.60 | 5.65 | 2.70 | 3.39 | 1.02 | 3.28 | 4.95 |
| PromptSRC | 1.54 | 4.89 | 7.69 | 15.36 | 1.48 | 1.72 | 5.85 | 2.81 | 1.88 | 0.99 | 2.39 | 4.24 |

(b) DAC

| Method | Caltech101 | DTD | EuroSAT | FGVC | Food101 | ImageNet | OF | OP | CARS | UCF | SUN | AVG |
|---|---|---|---|---|---|---|---|---|---|---|---|---|
| ZeroshotCLIP | 1.56 | 9.17 | 9.27 | 6.02 | 1.79 | 2.12 | 4.78 | 3.41 | 3.38 | 3.68 | 5.57 | 4.61 |
| CoCoOp | 1.43 | 5.96 | 9.18 | 3.84 | 3.80 | 7.85 | 5.96 | 3.72 | 6.32 | 8.28 | 5.22 | 5.60 |
| KgCoOp | 1.21 | 4.09 | 9.30 | 9.21 | 2.56 | 1.95 | 4.50 | 3.48 | 3.41 | 1.67 | 3.60 | 4.09 |
| MaPLe | 1.74 | 6.31 | 10.98 | 8.51 | 3.39 | 2.63 | 8.59 | 4.09 | 4.98 | 6.55 | 7.44 | 5.93 |
| ProGrad | 2.21 | 10.49 | 8.59 | 4.52 | 4.23 | 3.18 | 4.43 | 5.11 | 5.94 | 8.42 | 10.86 | 6.18 |
| PromptSRC | 1.40 | 4.33 | 9.67 | 12.37 | 2.57 | 1.66 | 5.34 | 3.36 | 2.45 | 2.85 | 2.56 | 4.41 |

(c) CAC

| Method | Caltech101 | DTD | EuroSAT | FGVC | Food101 | ImageNet | OF | OP | CARS | UCF | SUN | AVG |
|---|---|---|---|---|---|---|---|---|---|---|---|---|
| ZeroshotCLIP | 2.03 | 8.87 | 6.87 | 10.86 | 0.74 | 5.92 | 6.83 | 1.83 | 1.64 | 2.29 | 1.60 | 4.50 |
| CoCoOp | 1.82 | 7.92 | 9.6 | 8.43 | 1.65 | 1.62 | 4.9 | 3.3 | 2.68 | 1.2 | 3.27 | 4.22 |
| KgCoOp | 1.57 | 5.04 | 6.61 | 9.06 | 0.34 | 4.75 | 5.47 | 2.23 | 2.69 | 1.03 | 2.82 | 3.78 |
| MaPLe | 2.02 | 10.63 | 16.09 | 8.54 | 1.21 | 2.32 | 5.62 | 2.8 | 4.4 | 2.07 | 2.63 | 5.30 |
| ProGrad | 1.9 | 5.55 | 10.64 | 5.52 | 1.59 | 2.18 | 4.33 | 3.45 | 3.16 | 2.15 | 4.08 | 4.05 |
| PromptSRC | 1.66 | 5.04 | 7.42 | 7.7 | 0.84 | 2.18 | 5.7 | 2.08 | 1.67 | 1.32 | 1.83 | 3.40 |

Table 20: Maximum Calibration Error (MCE) on unseen classes in diverse downstream datasets using various tuning methods for CLIP-ViT-B/16.

(a) W/O Calibration

| Method | Caltech101 | DTD | EuroSAT | FGVC | Food101 | ImageNet | OF | OP | CARS | UCF | SUN | AVG |
|---|---|---|---|---|---|---|---|---|---|---|---|---|
| ZeroshotCLIP | 0.49 | 2.84 | 2.45 | 1.75 | 0.62 | 0.49 | 1.03 | 1.20 | 0.73 | 0.68 | 1.11 | 1.22 |
| CoCoOp | 1.01 | 2.88 | 3.10 | 2.81 | 0.59 | 0.43 | 1.52 | 0.82 | 0.51 | 0.51 | 0.97 | 1.38 |
| KgCoOp | 0.46 | 1.35 | 2.11 | 2.71 | 0.65 | 0.39 | 1.14 | 1.24 | 0.71 | 0.32 | 0.77 | 1.08 |
| MaPLe | 0.84 | 3.71 | 5.59 | 4.07 | 0.46 | 0.83 | 4.33 | 1.04 | 0.73 | 0.65 | 0.66 | 2.08 |
| ProGrad | 1.31 | 1.77 | 4.32 | 2.67 | 0.55 | 0.80 | 1.25 | 1.07 | 0.94 | 0.33 | 1.12 | 1.47 |
| PromptSRC | 0.66 | 1.11 | 2.29 | 3.56 | 0.56 | 0.56 | 1.42 | 1.00 | 0.52 | 0.29 | 0.78 | 1.16 |

(b) DAC

| Method | Caltech101 | DTD | EuroSAT | FGVC | Food101 | ImageNet | OF | OP | CARS | UCF | SUN | AVG |
|---|---|---|---|---|---|---|---|---|---|---|---|---|
| ZeroshotCLIP | 0.49 | 2.84 | 2.45 | 1.75 | 0.62 | 0.49 | 1.03 | 1.20 | 0.73 | 0.68 | 1.11 | 1.22 |
| CoCoOp | 0.87 | 1.68 | 3.34 | 0.79 | 1.09 | 1.17 | 1.35 | 1.25 | 1.39 | 1.30 | 1.15 | 1.40 |
| KgCoOp | 0.42 | 1.17 | 3.07 | 2.49 | 0.83 | 0.40 | 1.03 | 1.30 | 0.81 | 0.37 | 1.05 | 1.18 |
| MaPLe | 0.62 | 2.01 | 4.60 | 1.83 | 1.07 | 0.51 | 1.53 | 1.45 | 1.27 | 1.11 | 1.82 | 1.62 |
| ProGrad | 1.22 | 2.83 | 2.50 | 0.85 | 1.08 | 0.63 | 1.10 | 1.65 | 1.25 | 1.40 | 2.37 | 1.53 |
| PromptSRC | 0.60 | 1.15 | 2.75 | 2.89 | 0.87 | 0.40 | 1.16 | 1.19 | 0.56 | 0.58 | 0.76 | 1.17 |

(c) CAC

| Method | Caltech101 | DTD | EuroSAT | FGVC | Food101 | ImageNet | OF | OP | CARS | UCF | SUN | AVG |
|---|---|---|---|---|---|---|---|---|---|---|---|---|
| ZeroshotCLIP | 1.02 | 2.39 | 2.40 | 3.06 | 0.21 | 1.80 | 1.17 | 0.60 | 0.30 | 0.83 | 0.52 | 1.30 |
| CoCoOp | 0.91 | 1.73 | 3.17 | 2.39 | 0.55 | 0.41 | 0.96 | 1.28 | 0.67 | 0.32 | 0.78 | 1.20 |
| KgCoOp | 0.75 | 1.45 | 2.17 | 2.33 | 0.14 | 1.50 | 1.13 | 0.93 | 0.67 | 0.3 | 0.76 | 1.10 |
| MaPLe | 0.74 | 2.49 | 6.22 | 2.47 | 0.48 | 0.67 | 1.12 | 1.17 | 1.04 | 0.42 | 0.94 | 1.61 |
| ProGrad | 1.20 | 1.55 | 3.29 | 1.76 | 0.59 | 0.63 | 1.00 | 1.35 | 0.66 | 0.45 | 1.21 | 1.24 |
| PromptSRC | 0.68 | 1.14 | 2.29 | 2.52 | 0.31 | 0.82 | 1.26 | 0.78 | 0.52 | 0.33 | 0.65 | 1.03 |

Table 21: Proximity-Informed Expected Calibration Error (PIECE) on unseen classes in diverse downstream datasets using various tuning methods for CLIP-ViT-B/16.

(a) W/O Calibration

| Method | Caltech101 | DTD | EuroSAT | FGVC | Food101 | ImageNet | OF | OP | CARS | UCF | SUN | AVG |
|---|---|---|---|---|---|---|---|---|---|---|---|---|
| ZeroshotCLIP | 3.69 | 14.22 | 10.14 | 10.24 | 2.24 | 2.98 | 7.99 | 3.89 | 4.76 | 4.39 | 7.98 | 6.59 |
| CoCoOp | 4.28 | 16.24 | 13.04 | 15.58 | 2.26 | 2.59 | 9.13 | 3.08 | 4.65 | 3.25 | 6.78 | 7.35 |
| KgCoOp | 3.85 | 12.05 | 9.75 | 13.32 | 2.40 | 2.78 | 8.15 | 3.86 | 5.21 | 2.90 | 6.69 | 6.45 |
| MaPLe | 4.16 | 22.34 | 20.80 | 18.38 | 1.87 | 3.61 | 15.27 | 3.39 | 4.99 | 3.49 | 6.54 | 9.53 |
| ProGrad | 5.01 | 11.67 | 17.25 | 13.17 | 2.26 | 3.30 | 8.86 | 3.63 | 5.44 | 2.98 | 7.98 | 7.41 |
| PromptSRC | 3.55 | 12.94 | 10.80 | 16.51 | 2.08 | 2.72 | 8.21 | 3.83 | 4.17 | 2.74 | 6.13 | 6.70 |

(b) DAC

| Method | Caltech101 | DTD | EuroSAT | FGVC | Food101 | ImageNet | OF | OP | CARS | UCF | SUN | AVG |
|---|---|---|---|---|---|---|---|---|---|---|---|---|
| ZeroshotCLIP | 3.69 | 14.22 | 10.14 | 10.24 | 2.24 | 2.98 | 7.99 | 3.89 | 4.76 | 4.39 | 7.98 | 6.59 |
| CoCoOp | 4.81 | 12.37 | 12.34 | 8.68 | 4.06 | 7.86 | 9.56 | 4.43 | 7.73 | 8.40 | 8.43 | 8.06 |
| KgCoOp | 3.88 | 11.51 | 11.52 | 12.26 | 2.87 | 2.75 | 8.19 | 4.11 | 5.39 | 3.05 | 7.31 | 6.62 |
| MaPLe | 4.16 | 11.99 | 16.60 | 11.74 | 3.58 | 3.06 | 10.83 | 4.90 | 6.66 | 6.75 | 9.78 | 8.19 |
| ProGrad | 5.43 | 13.87 | 11.16 | 8.70 | 4.50 | 3.67 | 8.60 | 5.83 | 7.26 | 8.54 | 12.65 | 8.20 |
| PromptSRC | 3.58 | 12.46 | 12.22 | 14.20 | 2.91 | 2.57 | 7.95 | 4.29 | 4.78 | 3.78 | 6.24 | 6.82 |

(c) CAC

| Method | Caltech101 | DTD | EuroSAT | FGVC | Food101 | ImageNet | OF | OP | CARS | UCF | SUN | AVG |
|---|---|---|---|---|---|---|---|---|---|---|---|---|
| ZeroshotCLIP | 3.59 | 14.01 | 8.49 | 13.39 | 1.33 | 6.26 | 8.15 | 2.77 | 4.07 | 3.43 | 6.16 | 6.51 |
| CoCoOp | 4.32 | 13.56 | 12.6 | 12.56 | 2.15 | 2.66 | 7.81 | 4.13 | 5.05 | 2.86 | 7.4 | 6.83 |
| KgCoOp | 3.94 | 12.24 | 9.47 | 11.94 | 1.34 | 5.21 | 8.27 | 3.04 | 5.26 | 2.89 | 6.74 | 6.39 |
| MaPLe | 4.04 | 14.86 | 20.58 | 11.16 | 1.93 | 3.06 | 9.18 | 3.74 | 5.96 | 3.23 | 6.8 | 7.69 |
| ProGrad | 4.94 | 10.98 | 12.35 | 10.74 | 2.31 | 3.07 | 8.31 | 4.38 | 5.46 | 3.29 | 8.37 | 6.75 |
| PromptSRC | 3.62 | 12.92 | 10.75 | 10.16 | 1.66 | 3.16 | 8.19 | 3.33 | 4.38 | 2.84 | 6.31 | 6.12 |

Table 22: Detailed ECE and ACE results on cross-dataset setting (Conf and CAC)

(a) Original ECE results

| | Source | Target | | | | | | | | | | |
| | ImageNet | Cal | DTD | EuroSAT | FGVC | Food101 | OF | OP | CARS | SUN | UCF | AVG |
|---|---|---|---|---|---|---|---|---|---|---|---|---|
| CoCoOp | 2.28 | 4.26 | 5.32 | 5.67 | 3.59 | 2.17 | 2.61 | 5.36 | 8.21 | 2.97 | 2.61 | 4.28 |
| KgCoOp | 1.93 | 3.62 | 7.41 | 7.57 | 5.39 | 2.37 | 2.3 | 5.04 | 4.82 | 3.05 | 2.13 | 4.37 |
| MaPLe | 2.04 | 2.41 | 7.49 | 3.55 | 2.81 | 2.77 | 2.63 | 5.46 | 7.53 | 3.64 | 2.07 | 4.04 |
| ProGrad | 2.32 | 3.94 | 6.79 | 3.11 | 6.18 | 2.59 | 2.81 | 5.18 | 7.02 | 2.22 | 1.71 | 4.16 |
| PromptSRC | 2.11 | 3.73 | 6.67 | 7.82 | 3.55 | 2.09 | 2.68 | 4.63 | 6.89 | 3.58 | 2.34 | 4.40 |

(b) ECE results of CAC

| | Source | Target | | | | | | | | | | |
| | ImageNet | Cal | DTD | EuroSAT | FGVC | Food101 | OF | OP | CARS | SUN | UCF | AVG |
|---|---|---|---|---|---|---|---|---|---|---|---|---|
| CoCoOp | 1.36 | 3.56 | 7.45 | 6.52 | 4.6 | 1.26 | 2.43 | 4.5 | 6.79 | 1.64 | 2.77 | 4.15 |
| KgCoOp | 1.7 | 2.5 | 9.43 | 8.24 | 6.71 | 1.11 | 2.35 | 4.14 | 2.97 | 1.91 | 2.11 | 4.15 |
| MaPLe | 1.32 | 2.06 | 9.48 | 3.4 | 3.99 | 1.64 | 2.4 | 4.62 | 5.98 | 2.12 | 1.46 | 3.72 |
| ProGrad | 1.13 | 3.26 | 8.4 | 3.69 | 6.73 | 1.32 | 3.09 | 4.42 | 5.53 | 1.75 | 2.07 | 4.03 |
| PromptSRC | 1.26 | 2.98 | 8.76 | 8.87 | 4.91 | 0.94 | 1.64 | 3.68 | 4.94 | 1.79 | 0.68 | 3.92 |

(c) Original ACE results

| | Source | Target | | | | | | | | | | |
| | ImageNet | Cal | DTD | EuroSAT | FGVC | Food101 | OF | OP | CARS | SUN | UCF | AVG |
|---|---|---|---|---|---|---|---|---|---|---|---|---|
| CoCoOp | 2.28 | 4.26 | 6.14 | 6.51 | 3.42 | 2.13 | 2.92 | 4.74 | 8.15 | 3.02 | 2.65 | 4.39 |
| KgCoOp | 1.97 | 3.62 | 7.35 | 7.68 | 5.32 | 2.34 | 2.58 | 4.50 | 4.80 | 3.06 | 2.09 | 4.33 |
| MaPLe | 2.13 | 2.44 | 7.41 | 4.23 | 2.74 | 2.79 | 2.79 | 4.87 | 7.50 | 3.66 | 2.23 | 4.07 |
| ProGrad | 2.32 | 3.94 | 6.73 | 2.89 | 5.90 | 2.60 | 2.74 | 4.87 | 7.02 | 2.20 | 1.74 | 4.06 |
| PromptSRC | 2.14 | 3.57 | 6.84 | 7.88 | 3.51 | 2.11 | 2.63 | 4.31 | 6.87 | 3.48 | 2.12 | 4.33 |

(d) ACE results of CAC

| | Source | Target | | | | | | | | | | |
| | ImageNet | Cal | DTD | EuroSAT | FGVC | Food101 | OF | OP | CARS | SUN | UCF | AVG |
|---|---|---|---|---|---|---|---|---|---|---|---|---|
| CoCoOp | 1.44 | 3.56 | 7.70 | 7.00 | 4.12 | 1.40 | 2.84 | 3.88 | 6.71 | 1.93 | 2.33 | 4.15 |
| KgCoOp | 1.73 | 2.50 | 9.43 | 8.36 | 6.69 | 1.11 | 2.31 | 3.70 | 3.05 | 1.91 | 2.06 | 4.11 |
| MaPLe | 1.35 | 1.89 | 9.39 | 4.03 | 4.00 | 1.67 | 2.38 | 4.12 | 5.81 | 2.24 | 2.05 | 3.76 |
| ProGrad | 1.18 | 3.26 | 8.47 | 4.04 | 6.57 | 1.34 | 3.46 | 4.00 | 5.53 | 1.50 | 2.20 | 4.04 |
| PromptSRC | 1.24 | 2.74 | 8.82 | 8.88 | 4.62 | 0.90 | 1.94 | 3.61 | 4.83 | 1.86 | 1.10 | 3.93 |

## E.5 Detailed Results of Cross-Dataset Setting

This section showcases detailed results of calibration for the cross-dataset setting, aiming to illustrate that CAC facilitates cross-dataset calibration in current prompt tuning. For a comprehensive evaluation, we use four standard metrics to assess confidence calibration: Expected Calibration Error (ECE) (Guo et al., 2017a) and Adaptive Calibration Error (ACE) (Nixon et al., 2019). Detailed results are shown in Table 22.

## E.6 Detailed Results of Domain-Generalization Setting

This section showcases detailed results of calibration for the domain-generalization setting, aiming to illustrate that CAC facilitates domain-generalization calibration in current prompt tuning. For a comprehensive evaluation, we use four standard metrics to assess confidence calibration: Expected Calibration Error (ECE) (Guo et al., 2017a) and Adaptive Calibration Error (ACE) (Nixon et al., 2019). Detailed results are shown in Table 23.

Table 23: Detailed ECE and ACE results on domain-generalization setting (Conf and CAC)

(a) Original ECE results

| | Source | Target | | | | |
|---|---|---|---|---|---|---|
| | ImageNet | ImageNet-A | ImageNet-R | ImageNet-S | ImageNet-V2 | AVG |
| CoCoOp | 2.28 | 8.95 | 2.80 | 2.58 | 1.54 | 3.97 |
| KgCoOp | 1.93 | 9.06 | 3.03 | 4.18 | 1.68 | 4.49 |
| MaPLe | 2.04 | 8.38 | 3.52 | 2.72 | 1.79 | 4.10 |
| ProGrad | 2.32 | 9.96 | 2.43 | 3.53 | 1.82 | 4.44 |
| PromptSRC | 2.11 | 8.00 | 3.05 | 3.53 | 1.82 | 4.10 |

(b) ECE results of CAC

| | Source | Target | | | | |
|---|---|---|---|---|---|---|
| | ImageNet | ImageNet-A | ImageNet-R | ImageNet-S | ImageNet-V2 | AVG |
| CoCoOp | 1.36 | 6.61 | 4.40 | 0.87 | 1.61 | 3.37 |
| KgCoOp | 1.7 | 7.05 | 0.86 | 2.16 | 1.78 | 2.96 |
| MaPLe | 1.32 | 5.67 | 3.38 | 0.64 | 2.08 | 2.94 |
| ProGrad | 1.13 | 7.39 | 2.39 | 1.26 | 1.69 | 3.18 |
| PromptSRC | 1.26 | 6.99 | 3.96 | 2.85 | 1.77 | 3.89 |

(c) Original ACE results

| | Source | Target | | | | |
|---|---|---|---|---|---|---|
| | ImageNet | ImageNet-A | ImageNet-R | ImageNet-S | ImageNet-V2 | AVG |
| CoCoOp | 2.28 | 8.90 | 2.89 | 2.58 | 1.47 | 3.96 |
| KgCoOp | 1.97 | 8.99 | 3.08 | 4.17 | 1.86 | 4.53 |
| MaPLe | 2.13 | 8.39 | 3.56 | 2.71 | 2.07 | 4.18 |
| ProGrad | 2.32 | 9.96 | 2.51 | 3.53 | 1.54 | 4.39 |
| PromptSRC | 2.14 | 7.98 | 3.11 | 3.53 | 1.72 | 4.09 |

(d) ACE results of CAC

| | Source | Target | | | | |
|---|---|---|---|---|---|---|
| | ImageNet | ImageNet-A | ImageNet-R | ImageNet-S | ImageNet-V2 | AVG |
| CoCoOp | 1.44 | 6.52 | 4.45 | 0.79 | 1.61 | 3.34 |
| KgCoOp | 1.73 | 6.98 | 0.96 | 2.16 | 1.89 | 3.00 |
| MaPLe | 1.35 | 5.64 | 3.46 | 0.80 | 2.15 | 3.01 |
| ProGrad | 1.18 | 7.39 | 2.47 | 1.26 | 1.54 | 3.17 |
| PromptSRC | 1.24 | 6.97 | 4.03 | 2.85 | 1.92 | 3.94 |

