# OpenReview forum: "Contrast-Aware Calibration for Fine-Tuned CLIP: Leveraging Image-Text Alignment"
_ICLR.cc/2026/Conference — Submitted to ICLR 2026_

### Official Review · Reviewer_v9x2 · 2025-10-24

**Soundness:** 3
**Presentation:** 2
**Contribution:** 3
**Rating:** 4
**Confidence:** 5

**Summary:**

The paper introduces Contrast-Aware Calibration (CAC) for confidence calibration in fine-tuned CLIP.  The authors first point out that fine-tuned CLIP suffers from miscalibration and attribute miscalibration on unseen classes to weakened inter-class and intra-class discrimination, which is quantified by a “contrast” metric derived from logits. Motivated by this phenomenon, the authors propose CAC to rescale fine-tuned logits using a weight computed from the L1 distance between original CLIP and fine-tuned logits. Extensive experiments on 11 datasets show that CAC reduces calibration error without changing accuracy and with negligible overhead.

**Strengths:**

1.	The motivation is clear. The author presents a clear relation from contrast to calibration.
2.	The experimental results are promising. CAC effectively calibrates train classes, which outperform other calibration methods in Table 3. Moreover, A key strength is that CAC addresses a major limitation of previous DAC, which only applies to unseen classes.

**Weaknesses:**

1.	The connection between motivation and method seems weak. The paper invests significant effort to define a "Contrast Metric" in Section 3.1 and 3.2. However, the proposed method does not use this metric.
2.	The design of hyperparameters is heuristic, especially the piecewise thresholds. As shown in Table 5, CAC could show worse calibration than baseline without any hyperparameter. Such complexity weakens the contribution of CAC.
3.	The ablation for CAC is not enough. Is CAC sensitive to the metric for distance measurement in Equation 3?

**Questions:**

1.	Does the calibrated CLIP show a better degree of contrast?
2.	Can CAC be combined with training-based methods for better CLIP calibration [1-2]?

[1] Towards calibrated robust fine-tuning of vision-language models. NeurlPS, 2024.

[2] Understanding and Mitigating Miscalibration in Prompt Tuning for Vision-Language Models. ICML, 2025.

---

> ### Author Response · Authors · 2025-11-18
>
> We sincerely thank you for the positive and encouraging comments. Below, we provide detailed responses to each point raised.
>
> W1: The paper emphasizes a “Contrast Metric” in Sections 3.1–3.2, but the proposed method does not utilize it.
>
> A1: We are honored to address your query. It is important to clarify that **our proposed Contrast metric is dataset-level, which is computed via a similarity matrix $S \in \mathbb{R}^{N \times C}$ (where each row of the matrix corresponds to the Logits of a single sample)**. Thus, in practical computation, we only need to compare the Logits differences at the sample level to equivalently capture the dataset-level Contrast characteristics. Furthermore, as analyzed in Section 3.2, **the Contrast metric exhibits a significant negative correlation with the ECE**, and its calibration performance can be directly reflected by the ECE metric.
>
> W2: The heuristic hyperparameter design, particularly the piecewise thresholds, can sometimes lead CAC to perform worse than the baseline, weakening its contribution.
>
> A2: **We would like to clarify that the piecewise function is designed to handle special cases, serving as an enhancement rather than a necessity for CAC**. As shown in Table 5, even without the piecewise function, the standard CAC significantly reduces the ECE of the original model and achieves optimal or near-optimal results in most cases. Although performance may be slightly lower than the baseline DAC in a few rare instances, CAC remains highly competitive.  Furthermore, the parameter robustness of our method is reflected in its ability to adapt to different experimental settings—**including different datasets, model architectures, and shot numbers—using only a single set of parameters, while still achieving SOTA performance.** This again validates that our method is concise and effective compared to existing approaches.
>
> W3: Is CAC sensitive to the metric for distance measurement in Equation 3?
>
> A3: To address your question, we replaced the distance metric in Eq. 3 with the L2 distance while keeping all other parameters unchanged (**using the L14 architecture for KgCoOp and PromptSRC**). As shown in the table below, the performance of CAC changes only marginally. It is worth noting that this distance function is solely used to measure the logits difference between the original CLIP and the fine-tuned model, and various distance metrics can achieve this purpose. **We chose the L1 distance primarily for its computational simplicity and efficiency.**
>
>
> Q1: Does the calibrated CLIP show a better degree of contrast?
>
> A1: Your question is well-taken. We analyzed the contrast of two methods before and after calibration; the results are shown in the table below. As can be seen, CAC calibration indeed increases contrast. Moreover, as noted in the paper, **the original CLIP exhibits the highest contrast, which aligns with our conclusion that CLIP already possesses strong inherent calibration properties.**
>
>
> | Dataset | ZSCLIP | KgCoOp  | KgCoOp (calibrated) | PromptSRC | PromptSRC (calibrated) |
> | ------- | ------ | ------- | ------------------- | --------- | ---------------------- |
> | Food101 | 0.7691 | 0.7168  | 0.7444              | 0.6656    | 0.6723                 |
> | FGVC    | 0.0418 | -0.0610 | -0.0258             | -0.0048   | 0.0312                 |
>
>
> Q2: Can CAC be combined with training-based methods for better CLIP calibration [1-2]?
>
> A2: Similar to DAC, CAC is a post-hoc calibration module that decouples the fine-tuning process from the calibration mechanism: **the fine-tuning methods remain focused on improving recognition accuracy, while CAC can be applied afterward to enhance confidence calibration metrics such as ECE**. Therefore, from a methodological perspective, CAC can also be applied on top of training-based methods for better CLIP calibration. We will incorporate the discussion of papers in this direction into the revised version.
>
> [1] Towards calibrated robust fine-tuning of vision-language models. NeurlPS, 2024.
>
> [2] Understanding and Mitigating Miscalibration in Prompt Tuning for Vision-Language Models. ICML, 2025.

---

> ### Comment · Reviewer_v9x2 · 2025-11-25
>
> Thanks for the detailed rebuttal. Most concerns have been addressed. It is worth noting that CAC shows a better degree of contrast compared with baselines, which significantly supports the author's claim.
>
> However, I am still concerned about two things:
>
> 1.  **The motivation of contrast degree**: The motivation relies heavily on empirical observations of contrast degree. It remains unclear why the contrast degree should relate to calibration metrics (e.g., confidence).
> 2. **The sensitivity to hyperparameters**: The author argues that CAC is robust to hyperparameters. However, the results in Figure 5 show that ECE is not reduced in many cases, which needs further explanation.
>
> If the authors provide convincing resolutions to the above concerns, I will consider improving my score.

---

> > ### Author Response · Authors · 2025-11-26
> > **Official Comment by Authors**
> >
> > Dear reviewer v9x2：
> >
> > **To clarify why contrast can effectively reflect confidence calibration, we provide both theoretical justification and empirical evidence.**
> >
> > **Theoretical Perspective**:
> >
> > The model’s *confidence* is defined as the maximum predicted probability over all classes:
> > $$
> > \text{confidence} = \max_i(p_i)
> > $$
> >
> >
> >
> >
> > Let pos denote the logit of the ground-truth class and neg denote the largest logit among all negative classes. We define their difference as the *contrast*:
> > $$
> > m = e^{pos} - e^{neg}
> > $$
> >
> >
> > The softmax-normalized probability of the positive class is
> > $$
> > p = \frac{e^{pos}}{e^{pos} + e^{neg} + \sum_{i \in others} e^{i}}
> > $$
> > This probability admits the following upper bound:
> > $$
> > p = \frac{1}{1 + e^{-(pos - neg)} + \sum_{i \in others} e^{i - pos}} \le \frac{1}{1 + e^{-(pos - neg)}} = \frac{1}{1 + e^{-m}}
> > $$
> > Thus, **a larger contrast directly increases the upper bound of the predicted probability** p,
> >
> > implying a higher confidence level for the model.
> >
> >
> >
> > **Empirical Perspective**：As demonstrated in both the main paper and our previous rebuttal, extensive experiments across multiple datasets reveal a consistent **negative correlation** between contrast and ECE.
> >
> > Together with the above theoretical analysis, the empirical trends observed in your first reply, and the strong calibration performance achieved by CAC across diverse settings and datasets, we arrive at a unified conclusion:
> >
> > **Contrast is a reliable and theoretically grounded indicator of model confidence and calibration quality.** We hope this explanation fully addresses your concerns.
> >
> >
> > **Note**: We assume the mention of "Figure 5" refers to the parameter ablation studies or the sensitivity plots provided in the appendix.
> >
> > We claim robustness in the sense that a single set of default parameters ($k=15, \alpha=1.1$) achieves SOTA performance across 5 different fine-tuning methods, 11 diverse datasets, and 3 evaluation settings. Unlike traditional methods that require tuning per dataset, CAC works "out of the box."
> >
> > Explanation of ECE fluctuations:
> >
> > 1. Role of $k$: $k$ amplifies the subtle $L_1$ distance ($z$). If $k$ is set too low, the correction is negligible. If $k$ is set excessively high (e.g., $k=25$), the penalty becomes too aggressive, potentially over-correcting. However, the range $k \in [10, 15]$ consistently yields calibration improvements over the baseline for almost all methods.
> > 2. Role of $\alpha$: $\alpha$ allows CAC to handle underconfidence. If $\alpha=1.0$, we can only reduce confidence. Since some fine-tuned models are underconfident (e.g., MaPLe on some datasets), a value of $\alpha > 1.0$ is necessary. While extreme values (e.g., $\alpha=1.2$) can degrade performance, the sweet spot around $1.1$ is stable across architectures.
> >
> > In summary, while extreme parameter values can naturally degrade performance (as with any algorithm), the optimal region is wide and stable, allowing the default configuration to consistently outperform baselines without requiring dataset-specific tuning. We hope these explanations resolve your concerns regarding the theoretical grounding of contrast and the practical robustness of our hyperparameters. We respectfully request that you consider raising your score based on these clarifications.
> >
> > Best regards,
> >
> > The Authors

---

### Official Review · Reviewer_1KfC · 2025-10-27

**Soundness:** 3
**Presentation:** 3
**Contribution:** 2
**Rating:** 4
**Confidence:** 2

**Summary:**

This paper proposes a contrastive perception calibration method to address the issue of poor confidence calibration on unseen categories in fine-tuned CLIP models. The method leverages the discrepancy between the outputs of the original and fine-tuned CLIP models to design a training-free post-hoc calibration strategy, which improves calibration performance across various settings, including open-vocabulary recognition, cross-dataset evaluation, and domain generalization, without compromising model accuracy.

**Strengths:**

1. The overall presentation is clear and easy to follow;

2. The proposed plug-and-play component can be seamlessly integrated into vision-language models;

3. The method achieves promising results across multiple benchmarks and evaluation settings;

**Weaknesses:**

1. The use of an exponential function for weight adjustment is largely based on empirical observation and engineering heuristics, lacking deeper exploration of the function space or theoretical motivation. In fact, there exist numerous weight adjustment and parameter update mechanisms, especially within optimizers.

2. Exploration of intra-class and inter-class relationships has been studied in many prior works. While the authors implement this from a confidence-based perspective, the strategy is, in essence, similar to uncertainty-aware or uncertainty-estimation approaches.

3. The authors emphasize that CAC is insensitive to hyperparameters and adopt default values (k=15, α=1.10). However, the ablation studies in the appendix (e.g., Tables 10 and 11) indicate that performance does fluctuate with different values of k and α. For instance, when k=5 or 25, the ECE results deteriorate noticeably. Therefore, the claim of “hyperparameter insensitivity” may need to be stated more cautiously.

4. This work primarily focuses on CLIP and its prompt-learning variants. However, the development of vision-language models has now entered the era of multimodal large language models, such as LLaVA. The paper could broaden its perspective in the conclusion and future work by discussing whether and how the CAC approach, which leverages the alignment capabilities of a foundation model, might be applicable to these more complex, generative model architectures.

**Questions:**

Please refer to the Weaknesses part.

---

> ### Author Response · Authors · 2025-11-18
>
> Thank you for your comment. We will respond to each of the questions you raised.
>
> W1: The lack of analysis in the use of exponential functions may affect parameter optimization.
>
> A1: We would like to clarify that CAC is a **post-hoc method** that requires no training and **does not interfere with the prompt tuning process of the original model**. Therefore, the choice of the calibration function does not affect the optimizer’s parameters. We have discussed the theoretical motivation for using $e^{-x}$ in Section 4.1, and the ablation study in Table 5 empirically validates the effectiveness of $e^{-x}$ by comparing it with other decreasing functions.
>
> W2: Exploration of intra-class and inter-class relationships has been studied in many prior works. While the authors implement this from a confidence-based perspective, the strategy is, in essence, similar to uncertainty-aware or uncertainty-estimation approaches.
>
> A2: To the best of our knowledge, our work is **the first to analyze model confidence and perform calibration by explicitly leveraging inter-class and intra-class relationships**. We would appreciate it if you could specify which prior works you are referring to, so that we can include them in our discussion. In addition, our method achieves sample-level confidence calibration by contrasting the Contrast differences between the fine-tuned model and the original CLIP. While it aligns with the core idea of "uncertainty-aware/estimation approaches," there exists a significant distinction in the implementation pathway (i.e., the source of confidence calibration). Essentially, both approaches fall under the framework of "estimating confidence by quantifying the model's reliability (confidence/uncertainty) for each sample." Specifically, our CAC method centers on "contrasting the Contrast differences between the fine-tuned model and the original CLIP," whereas traditional uncertainty-aware/estimation methods rely more on "the model's own prediction probabilities, ensemble discrepancies, or Bayesian parameter distributions." We would appreciate a more detailed explanation of this issue and welcome further discussions with you. We would be grateful if you could provide more detailed feedback on this point for further discussion.
>
> W3: Although CAC uses default hyperparameters, performance varies with different values of k and $\alpha$, so the claim of hyperparameter insensitivity should be stated more cautiously.
>
> A3: We would like to clarify that **our statement regarding hyperparameter insensitivity means that CAC achieves SOTA performance across various settings and datasets using a single set of default parameters**. For example, using the default values of $\alpha$ = 1.1 and k=15, CAC achieves strong calibration even on datasets where DAC fails to calibrate effectively (see the table below).  As shown in Tables 10–11, we determine the optimal combination of $\alpha$ and k as the default setting, and subsequent ablation studies demonstrate the robustness of these hyperparameters across different experimental setups, datasets, model architectures, and shot numbers. The fact that **CAC consistently achieves strong performance under these default parameters highlights the design stability of our method.**
>
> | Method    | Origin | DAC  | CAC      |
> | --------- | ------ | ---- | -------- |
> | CoCoOp    | 5.44   | 5.70 | **4.24** |
> | KgCoOp    | 3.98   | 4.11 | **3.85** |
> | PromptSRC | 4.29   | 4.55 | **3.47** |
>
> W4: The paper focuses on CLIP and prompt-learning variants but should discuss CAC’s potential applicability to multimodal large language models like LLaVA.
>
> A4: Regarding the generalization beyond the CLIP family, we conducted a comprehensive review of miscalibration issues in other VLMs and found that miscalibration is a common and widespread problem. In fact, Tu et al. [1] evaluated 35 different VLMs and observed that all exhibited confidence calibration issues. As for LLaVA-type multimodal large language models, we note that **recent reinforcement learning approaches such as GRPO employ a reference model’s logits to adjust the fine-tuned model’s output distribution**. This idea is conceptually similar to CAC, which calibrates outputs using the original model to mitigate overfitting. Based on our in-depth analysis and sota experimental results, we conclude that CAC demonstrates strong confidence calibration capability, **with default hyperparameters showing robust performance across diverse datasets and evaluation settings**.
>
> [1] Tu W, Deng W, Campbell D, et al. An empirical study into what matters for calibrating vision-language models. ICML 2024.

---

> ### Comment · Reviewer_1KfC · 2025-11-26
>
> Thank you for the authors’ response. Class-centric exploration has been investigated in many prior works; I would even consider it a common understanding among researchers in multimodal/vision fields. However, the authors claim that this is the first attempt in the relevant area. I briefly list several related works, although they focus on downstream tasks [1,2,3]. Taking into account the evaluations from the other reviewers, I have lowered my score.
>
> [1]. What Makes CLIP More Robust to Long-Tailed Pre-Training Data? A Controlled Study for Transferable Insights, NeurIPS 2024
>
> [2]. CorrCLIP: Reconstructing Patch Correlations in CLIP for Open-Vocabulary Semantic Segmentation, ICCV 2025
>
> [3]. Multi-label Cluster Discrimination for Visual Representation Learning, ECCV 2024

---

> > ### Author Response · Authors · 2025-11-26
> >
> > Dear Reviewer:
> >
> > Thank you for providing the three relevant papers. We have carefully studied these works and found that their research focus fundamentally differs from ours:
> >
> > 1. Paper [1] (NeurIPS 2024): The core focus is on enhancing CLIP's robustness under long-tailed pre-training data by analyzing how data distribution affects model generalization capabilities. It addresses the problem of "performance degradation caused by imbalanced data distribution" and does not involve any discussion or optimization related to confidence calibration.
> >
> > 2. Paper [2] (ICCV 2025): This work concentrates on open-vocabulary semantic segmentation tasks, reducing inter-class interference by reconstructing correlations between image patches. Its primary goal is to improve segmentation accuracy and spatial consistency, with no connection to the calibration of model output confidence.
> >
> > 3. Paper [3] (ECCV 2024): This paper proposes a multi-label cluster discrimination method (MLCD) that enhances visual representation learning by assigning multiple pseudo-labels to images and designing specialized loss functions. It ultimately serves to improve performance on downstream tasks such as linear probing and zero-shot classification, but similarly does not address confidence calibration issues.
> >
> > Our work fundamentally differs from the aforementioned literature: we specifically focus on the problem of **confidence calibration**. What we claim as "first" refers to being the first to propose achieving sample-level confidence calibration by explicitly leveraging inter-class and intra-class relationships through contrasting the differences between the fine-tuned model and the original CLIP model. No existing literature within the confidence calibration framework employs this contrast-difference-based calibration mechanism. We are indeed the first work in this direction, and this innovation is thoroughly validated by our experimental results—our method significantly outperforms existing calibration methods across multiple benchmarks. Therefore, CAC is the first work that explicitly applies the inter-class/intra-class relationship contrast mechanism to confidence calibration in VLMs. We are also willing to more precisely articulate the distinctions from related works in our paper.
> >
> > Best regards,
> >
> > The Authors

---

### Official Review · Reviewer_wsjC · 2025-10-30

**Soundness:** 4
**Presentation:** 4
**Contribution:** 3
**Rating:** 6
**Confidence:** 4

**Summary:**

This paper targets the over-confidence problem of fine-tuned CLIP on unseen classes in open-vocabulary scenarios. It proposes a training-free, plug-and-play multimodal post-processing method called Contrast-Aware Calibration (CAC). The core idea is to treat the logit-level contrastive difference between the original CLIP and the fine-tuned CLIP as a miscalibration score, transform it into sample-specific scaling weights via an exponential function, and re-scale the temperature of the fine-tuned model accordingly. In this way, CAC simultaneously calibrates both training and unseen classes.

**Strengths:**

1. Originality: For the first time, the paper uses the “contrastive difference” itself as a direct miscalibration measure and establishes its negative correlation with ECE; it achieves sample-level calibration without any training data or validation set.
2. Quality: The derivation is rigorous, and ablation studies are comprehensive.
3. Clarity: The paper is well-structured; figures and tables intuitively verify the effectiveness of the proposed method.
4. Significance: It removes the limitation that existing VLM calibration methods can only handle either training classes or in-distribution unseen classes, providing a unified and trustworthy confidence output for open-vocabulary deployment.

**Weaknesses:**

1. The method seems to assume that the original CLIP is well-calibrated. If the pre-trained model itself has systematic bias, CAC may amplify this bias.
2. Experiments are conducted only on ViT-B/16, ViT-B/32, and RN50; larger backbones such as ViT-L/14 or other VLMs are not covered.
3. End-to-end evaluations in concrete application scenarios are missing; only classification ECE is reported.
4. Although the default parameters work well, the involved hyper-parameters still appear to be manually tuned, and no automatic selection strategy is provided.

**Questions:**

1. If the original CLIP itself is under-calibrated on certain domains, will CAC pull the fine-tuned model toward even worse confidence?
2. Performance varies greatly across different CLIP sizes; could you provide comparative results on larger CLIP variants?

---

> ### Author Response · Authors · 2025-11-18
>
> We sincerely thank you for the positive and encouraging comments. Below are our detailed responses.
>
> W1 & Q1: Concern about whether CAC may worsen calibration when the original CLIP is itself under-calibrated in certain domains.
>
> A1: Your concern is understandable. However, extensive research [1,2,3] has demonstrated that **CLIP’s pretraining leads to good calibration across multiple tasks**, whereas fine-tuning often disrupts modality alignment and causes severe confidence miscalibration. Therefore, in most cases, we can rely on CLIP’s original calibration performance. If CLIP itself exhibits calibration imbalance for certain downstream datasets (e.g., overconfidence in some classes, overall underconfidence), **CAC’s design also can ensure effective calibration** by referencing differences between model outputs. It adaptively provides conservative corrections for both underconfident and overconfident cases on a per-sample basis, which explains why CAC does not cause miscalibration. Across 11 datasets, we observed consistent improvement with CAC.
>
> W2 & Q2: The experiments evaluate only small/medium CLIP backbones (ViT-B/16, ViT-B/32, RN50), without covering larger models such as ViT-L/14 or other VLMs.
>
> A2: We appreciate the question. Table 6 in the main text already reports the performance differences between DAC and CAC across common CLIP architectures. **CAC outperforms DAC across all datasets and all architectures, establishing it as the current SOTA calibration method.**
>
> To further address your concern, we additionally evaluate CAC and DAC on the ViT-L/14 CLIP model. The results are shown below, demonstrating that **CAC is insensitive to backbone changes** and maintains stability across datasets and settings.
>
> |Method|ZSCLIP|KgCoOp (ECE)|KgCoOp (ACE)|PromptSRC (ECE)|PromptSRC (ACE)|
> |-|-|-|-|-|-|
> |Origin|2.85|3.79|3.74|4.36|4.24|
> |DAC|-|3.90|3.87|3.98|4.02|
> |CAC|-|3.55|3.55|3.52|3.32|
>
> W3: End-to-end evaluations in concrete application scenarios are missing; only classification ECE is reported.
>
> A3: Following DAC and standard CLIP-tuning protocols, **our goal is calibration improvement without harming accuracy**. Following established practice, we adopt ECE as the main metric and also report ACE, MCE, and PIECE.
>
> Although we do not include full end-to-end application scenarios, our experiments demonstrate that the method is robust across different experimental settings, datasets, model architectures, and shot numbers, achieving SOTA performance—indicating strong applicability.
>
> Additionally, to further test generality in specialized domains, we evaluate on **three representative remote-sensing datasets**. As shown below, CAC consistently improves unseen-class calibration for PromptSRC, confirming broad applicability in challenging scenarios.
>
> |Dataset|Origin|TS|DAC|CAC|
> |-|-|-|-|-|
> |PATTERNNET|2.66|9.10|3.17|2.50|
> |RESISIC45|7.71|12.27|7.93|6.83|
> |WHURS19|6.48|3.17|6.07|3.06|
>
> W4: Concern that CAC still relies on manual tuning and lacks an automatic hyper-parameter selection strategy.
>
> A4: For a training-free method, hyper-parameter stability is one of CAC’s key advantages. Even on unseen datasets, the default parameters already yield strong performance, meaning no tuning is required for initial use. For example, using the default values of $\alpha$ = 1.1 and k=15, CAC achieves strong calibration even on datasets where DAC fails to calibrate effectively (see the table below).
>
> Furthermore, **the roles of each parameter are elaborated in the main text**: Parameter $\alpha$ serves as the core regulator for confidence calibration, where $\alpha$ is increased for underconfident models and decreased for overconfident ones to directly correct confidence biases; Parameter k functions as an input magnitude scaling factor, primarily adapting to the input specifications of different baseline models or evaluation settings without altering the core calibration logic; The default parameters are selected based on the principles of universality and optimal calibration performance, validated through extensive experiments across 11 datasets and 5 fine-tuning methods; For most fine-tuning tasks of VLMs, the default parameters are sufficient; **If parameter adjustment is required, hyperparameters can be quickly determined by testing on a single dataset following our provided Parameter Selection Guidelines.** While automatic grid search is also a viable option, it is notably less efficient than our proposed Parameter Selection Guidelines.
>
> |Method|Origin|DAC|CAC|
> |-|-|-|-|
> |CoCoOp|5.44|5.70|**4.24**|
> |KgCoOp|3.98|4.11|**3.85**|
> |PromptSRC|4.29|4.55|**3.47**|
>
> [1] Wang S, Wang J, Wang G, et al. Open-vocabulary calibration for fine-tuned CLIP. ICML 2024.
>
> [2] Minderer M, Djolonga J, Romijnders R, et al. Revisiting the calibration of modern neural networks. NeurIPS 2021.
>
> [3] Wang S, Li Y, Wei H. Understanding and mitigating miscalibration in prompt tuning for vision-language models. ICML 2025.

---

> > ### Comment · Reviewer_wsjC · 2025-11-26
> >
> > Thank you for your response. The authors' rebuttal has largely alleviated most of my primary concerns. While I believe this work makes a valuable contribution, the limitations mentioned in the other reviewers' comments lead me to conclude that there is no compelling reason to raise my score. Therefore, I will maintain my score.

---

### Official Review · Reviewer_MHoE · 2025-10-31

**Soundness:** 2
**Presentation:** 1
**Contribution:** 2
**Rating:** 2
**Confidence:** 3

**Summary:**

This paper introduces a calibration method called Contrast-Aware Calibration (CAC) for CLIP models. Specifically, CAC calculates the confidence difference between the original and fine-tuned CLIP models, which is then scaled and transformed using an exponential function. The resulting CAC score is employed to reweight the logits for improved calibration. Experimental results demonstrate that CAC enhances the performance of various fine-tuned CLIP models.

**Strengths:**

1. The paper presents extensive experimental results across different prompt-learning methods and datasets, as detailed in Appendix E.

2. Experimental results demonstrate the effectiveness of proposed method in multiple datasets.

**Weaknesses:**

1.  The claim is not well supported with evidence. For instance, the authors claim that over-confidence and under-confidence arise from high or low inter-class and intra-class similarity (Lines 187-197). However, Figure 1(a) does not show the inter-class or intra-class similarity across different datasets. It only illustrates the phenomenon of over-confidence and under-confidence on two specific datasets. Furthermore, the authors claim that contrast and ECE exhibit a negative correlation for unseen classes (Line 202). It’s unclear how this conclusion follows from Figure 1(c-d). Did the authors conduct experiments exclusively on unseen classes for these figures? This point needs clarification.

2.  In Eq (3), z represents the confidence difference between the original and fine-tuned VLMs. However, the relationship between this equation and the earlier claim that over-confidence and under-confidence arise from high or low inter-class/intra-class similarity is not well explained. A more detailed explanation of this connection would help clarify the authors' argument.

3.  In Lines 242-243, the authors state that "the logits of … are equivalent to the contrast metric." However, logits are vectors in the N-way classification problem, whereas the contrast metric in Eq (1) is a constant. The illustration is not entirely accurate and needs refinement for consistency.

4.  The performance of the proposed CAC and advanced CAC methods is highly sensitive to the values of hyper-parameters. As shown in Tables 11-12 (Appendix D), small variations in hyper-parameters can significantly impact performance. Additionally, CAC (and advanced CAC) uses four different hyper-parameters (k, $\alpha$, $\lambda_1$, $\lambda_2$), which limits the method's general applicability.

**Questions:**

See my questions in weakness

---

> ### Author Response · Authors · 2025-11-18
>
> Thank you for acknowledging our experimental results. We realize that there is a slight misunderstanding regarding the motivation and logical flow of our method, and we would like to clarify it here.
>
> W1.1: The claim that over-confidence and under-confidence arise from high or low inter-class and intra-class similarity is not well supported with evidence.
>
> A1.1: Your concern directly relates to the central conclusion of our work: **poor class discrimination ability is the underlying cause of confidence imbalance.** We selected two representative datasets, FGVC and Food101, whose **intra-class and inter-class similarity characteristics are clear**, making them ideal for analyzing dataset properties and the resulting confidence imbalance. From the main text, the original descriptions of the datasets[1,2], the t-SNE visualizations in Appendix C, and the quantitative results in the table below, we can observe the following. The intra-class and inter-class similarity statistics were computed as follows: we extracted image features using the CLIP ViT-B/16 model and calculated cosine similarities for both intra-class and inter-class pairs. For each dataset, we randomly selected 50 classes and sampled 20 images per class. We introduce ImageNet—a well-structured dataset with balanced category distribution and stable image quality—as a neutral reference baseline.
> - For FGVCAircraft, the inter-class similarity is 0.7019, compared to 0.6473 for ImageNet. Although the numerical difference appears small, such a gap is statistically meaningful when averaged over a large number of class pairs. Therefore, this strongly supports the statement that *“FGVC exhibits high inter-class similarity.”* For this dataset, the model tends to misclassify a large portion of samples into the same class (over-confidence: the model assigns overwhelmingly high probability to a single class even when it is incorrect, resulting in a negative contrast).
> - For Food101, the intra-class similarity is 0.7691, whereas ImageNet’s intra-class similarity is 0.8120. The lower intra-class similarity of Food101 implies higher intra-class variation. Consequently, on this dataset the model often produces correct classifications but with low confidence (under-confidence: low intra-class similarity makes the model uncertain, leading to multiple classes receiving relatively high probabilities, and the contrast approaches zero).
>
> From the trends in Figures 1(c–d), together with the strong calibration improvements achieved by CAC (**Origin: 4.29 → DAC: 4.55 → CAC: 3.47**), it is evident that similar phenomena occur across other datasets as well. In summary, **when a model cannot effectively distinguish the classes within a dataset, its intra-class and inter-class similarities become highly entangled, which leads to over-confidence or under-confidence.**
>
> |Dataset|Intra-class similarity|Inter-class similarity|
> |-|-|-|
> |FGVC|0.7828|0.7019|
> |Food101|0.7691|0.6469|
> |ImageNet|0.8120|0.6473|
>
> W1.2: Evidence for the negative correlation between contrast and ECE in Figure 1(c-d)
>
> A1.2: There exists a negative correlation between contrast and ECE—models with stronger class discriminability tend to exhibit better confidence calibration. Figures 1(c–d) visualize the contrast and ECE on **unseen classes**, and one can clearly observe the pattern that *higher ECE corresponds to lower contrast* (as indicated by the arrows). **For seen classes, the model’s class discriminability (captured by contrast) is affected by fine-tuning and thus no longer reflects the dataset’s intrinsic structure. Therefore, we focus visualization on unseen classes.** Nevertheless, the fact that CAC also calibrates the seen classes effectively and achieves SOTA performance among training-free methods further demonstrates that contrast reliably reflects confidence miscalibration.

---

> > ### Author Response · Authors · 2025-11-18
> >
> > W2: Connection between Eq. 3 and inter-class/intra-class similarity
> >
> > A2: Section 3.2, together with our responses to W1.1 and W1.2, explains that poor class discriminability—i.e., the model’s inability to adequately distinguish intra-class and inter-class differences—ultimately leads to over- and under-confidence. As noted, **the contrast computed in Eq. 1 quantifies the model’s class discriminability and thereby reflects the degree of confidence miscalibration**. However, since ground-truth labels are unavailable at test time, a well-calibrated reference model is required. **CLIP naturally satisfies this requirement—its strong intrinsic calibration is consistently supported by both our experimental results(see the table below) and prior studies[3,4,5]**. Based on this, we use the metric z to measure the discrepancy between the fine-tuned model’s logits and those of the original CLIP, allowing us to determine whether the fine-tuned model’s contrast has fallen below an appropriate level and thus perform calibration accordingly. This design elegantly resolves the challenge posed by the absence of ground-truth labels during testing, enabling an efficient and reliable confidence calibration solution.
> >
> > |Dataset|ZSCLIP|KgCoOp|KgCoOp (calibrated)|PromptSRC|PromptSRC (calibrated)|
> > |-|-|-|-|-|-|
> > |Food101|0.7691|0.7168|0.7444|0.6656|0.6723|
> > |FGVC|0.0418|-0.0610|-0.0258|-0.0048|0.0312|
> >
> > W3: Relationship between logits and the contrast metric (logits are vectors in the N-way classification problem, whereas the contrast metric in Eq. 1 is a constant)
> >
> > A3: Your understanding is correct—**this part concerns the practical implementation details of CAC**. In our paper, contrast is a scalar metric that reflects a model’s dataset-level discriminative ability and, according to the established pattern, can indicate its confidence calibration to some extent. Logits, by contrast, are $1 \times C$ vectors. In Eq. 1, **the similarity matrix $S \in \mathbb{R}^{N \times C}$ used to compute contrast is formed by stacking the logits of each sample. When a model possesses strong discriminative ability, its logits exhibit high probability for the true class and low probabilities for others, which corresponds to a high contrast value at the dataset level.** Therefore, in the practical implementation of CAC, we only need to examine the difference between the logits of the fine-tuned model and those of the original CLIP to achieve the contrast-aware calibration described by CAC.
> >
> > We hope this clarifies the underlying mechanism. Based on this principle, **CAC demonstrates strong calibration performance across unseen classes, cross-dataset evaluations, and domain generalization settings. Notably, it provides effective calibration even for PromptSRC—where DAC fails—achieving ECE improvements (Observe the calibration performance of DAC and CAC in the following table).** These extensive experimental results further support both the validity of our summarized pattern and the effectiveness of CAC.
> >
> > W4: Hyper-parameter sensitivity and general applicability
> >
> > A4: In our paper, **the default parameters already yield SOTA performance across various settings and datasets**. For example, using the default values of $\alpha$ = 1.1 and k=15, CAC achieves strong calibration even on datasets where DAC fails to calibrate effectively (see the table below). Tables 11–12 are merely ablation studies aimed at identifying generally optimal ranges for the hyper-parameters. Moreover, **the piecewise squared function is only an additional component of advanced CAC that can further improve performance, but it is not strictly required**. In practice, only two hyper-parameters—$\alpha$ and k—need to be set. Based on Tables 10–11, we selected a single effective configuration of $\alpha$ and k, and all subsequent experiments consistently use this setting to demonstrate the robustness of CAC across different experimental setups, datasets, model architectures, and shot numbers. This in itself **highlights the design stability of our hyper-parameters**.
> >
> > In rare extreme cases, the Parameter Selection Guidelines allow users to efficiently determine appropriate hyper-parameters with testing on only a single dataset.
> >
> > |Method|Origin|DAC|CAC|
> > |-|-|-|-|
> > | CoCoOp    | 5.44   | 5.70 | **4.24** |
> > | KgCoOp    | 3.98   | 4.11 | **3.85** |
> > | PromptSRC | 4.29   | 4.55 | **3.47** |
> >
> > [1] Maji S, Rahtu E, et al. Fine-grained visual classification of aircraft. CVPR 2013.
> >
> > [2] Bossard L, et al. Food-101–mining discriminative components with random forests. ECCV 2014.
> >
> > [3] Wang S, Wang J, Wang G, et al. Open-vocabulary calibration for fine-tuned CLIP. ICML 2024.
> >
> > [4] Minderer M, Djolonga J, Romijnders R, et al. Revisiting the calibration of modern neural networks. NeurIPS 2021.
> >
> > [5] Wang S, Li Y, Wei H. Understanding and mitigating miscalibration in prompt tuning for vision-language models. ICML 2025.

---

### Author Response · Authors · 2025-12-01
**Final Remarks by Authors**

Dear SAC, AC, and Reviewers,

We hope this message finds you well.

Given the recent re-assignment of area chairs and the reset of review scores following the OpenReview incident, we provide this summary comment to help the newly assigned AC fairly assess our submission.

We thank all reviewers and the AC for their constructive feedback and the earlier discussion. Through additional experiments and thorough analyses, we have now addressed the reviewers’ concerns regarding the choice of model framework, parameter sensitivity, and the relevance of the contrast-based metric. Specifically, our expanded experiments demonstrate that the calibration behavior of CAC remains robust across different model sizes, datasets, and fine-tuning variants.

Meanwhile, based on reviewers’ comments and the prior discussion, we summarize several key advantages of CAC:

- The design of CAC is sound, requires no training or test-time adaptation, and achieves SOTA performance. (Reviewers MHoE, v9x2)

- The motivation behind CAC is original—it is the first to jointly analyze confidence calibration and intra-/inter-class discriminative ability. (Reviewer wsjC)

- By using the original CLIP as a calibration anchor, CAC effectively resolves the limitation of prior methods that could only handle novel classes. (Reviewers v9x2, 1KfC)

Due to unforeseen circumstances, some discussions did not receive final responses. The key points are summarized as follows:

- Reviewer MHoE: We provided detailed clarification regarding their doubts about the motivation. Our response to reviewer wsjC further illustrates the relationship between contrast and confidence metrics.

- Reviewer 1KfC: He listed other works involving intra-/inter-class analysis. However, after careful examination, we found that the problems we address are fundamentally different from those works, and our motivations and analytical approaches also differ substantially. CAC is the first to explicitly leverage the difference between the fine-tuned model and the original CLIP to achieve sample-level confidence calibration through inter- and intra-class relational comparison. Under current confidence calibration frameworks, no prior literature has adopted such a differential contrast-based calibration mechanism.

- Reviewer v9x2: We addressed his concerns regarding the connection between contrast and confidence calibration from both theoretical derivation and empirical evidence. Based on our results, we showed that CAC, within an optimal parameter range, is capable of generalizing across datasets, methods, and experimental setups, thereby responding to their question on parameter sensitivity.

Overall, to the best of our knowledge, CAC is the first method capable of achieving confidence calibration across diverse settings, providing a valuable reference for future research. We sincerely hope the new AC will take into account the original reviews, our author responses, the prior discussion, and this summary comment, before making a recommendation.

We sincerely appreciate your time and efforts throughout the review process.

Best regards,

The Authors of Submission 8719

ICLR 2026

---

### Meta-Review · Area_Chair_X6Hi · 2026-01-06

**Summary:**

1. Weak Theoretical Grounding and Heuristic Nature: Multiple reviewers (MHoE, 1KfC, v9x2) criticized the lack of rigorous theoretical motivation for the core contrast-aware mechanism. The exponential weighting function and piecewise thresholds were described as empirical heuristics rather than derived principles, lacking exploration of alternative functions or deeper links to prior uncertainty-aware methods.
2. Hyperparameter Sensitivity and Practical Applicability: A recurring issue (MHoE, wsjC, 1KfC, v9x2) was the method's reliance on multiple hyperparameters (e.g., k, α, thresholds), with ablations showing performance fluctuations or drops in certain cases.
3. Limited Scope and Experimental Validation: Reviewers (wsjC, 1KfC) pointed out restrictions to smaller CLIP backbones (ViT-B/16, ViT-B/32, RN50), with calls for testing on larger models like ViT-L/14 or multimodal LLMs (e.g., LLaVA).

**Reviewer Concerns:**

Addressed Concerns:

1. Assumptions About Original CLIP Calibration (wsjC): Justified with references to prior studies showing CLIP's good pretraining calibration; emphasized CAC's per-sample corrections to avoid bias amplification, supported by consistent improvements on 11 datasets.

2. Insufficient Ablations (v9x2, partially 1KfC): Included ablations on alternative weighting functions (Table 5), distance metrics (L2 vs. L1, with marginal differences), and hyperparameters (Tables 10-12), showing L1's efficiency and optional piecewise thresholds.

3. Limited Experimental Scope (wsjC, partially 1KfC): Added results on larger backbones like ViT-L/14.

Outstanding Concerns:

1. Lack of Rigorous Theoretical Motivation and Heuristic Nature: Reviewers found the exponential weighting (e^{-x}) and contrast mechanism still largely empirical/heuristic, with insufficient derivation or exploration of function spaces.

2. Novelty and Differentiation from Prior Works (1KfC): The authors listed priors (e.g., NeurIPS 2024 on long-tailed data, ICCV 2025, ECCV 2024), but 1KfC deemed them unrelated (no direct calibration focus), weakening claims of being the "first" contrast-based sample-level calibration. Differentiation from uncertainty methods (e.g., no ensembles) was attempted but not convincing enough, leading to the score drop.

3. Residual Hyperparameter Sensitivity and Practical Limitations: While defaults were shown robust, v9x2 highlighted ongoing ECE fluctuations in ablations/Figure 5, and the absence of an automatic tuning strategy (wsjC) was not addressed. 1KfC viewed insensitivity claims as still overstated given performance drops at non-optimal values.

4. Broader End-to-End Applications (wsjC): The rebuttal followed DAC protocols and added metrics (ACE/MCE/PIECE), but lacked evaluations in full VLM tasks beyond classification ECE, leaving this partially open.

**Reviewer Scores:**

The reviewer MHoE may not increase the score, while wsjc maintains the score. The other two reviewers may increase to borderline scores.

---

### Decision · Program_Chairs · 2026-01-26

Reject